# Over half of western United States' most abundant tree species in decline

Hunter Stanke [1,2 ✉], Andrew O. Finley [1,3], Grant M. Domke [4], Aaron S. Weed[5] & David W. MacFarlane[1]

Changing forest disturbance regimes and climate are driving accelerated tree mortality across temperate forests. However, it remains unknown if elevated mortality has induced decline of tree populations and the ecological, economic, and social benefits they provide. Here, we develop a standardized forest demographic index and use it to quantify trends in tree population dynamics over the last two decades in the western United States. The rate and pattern of change we observe across species and tree size-distributions is alarming and often undesirable. We observe significant population decline in a majority of species examined, show decline was particularly severe, albeit size-dependent, among subalpine tree species, and provide evidence of widespread shifts in the size-structure of montane forests. Our findings offer a stark warning of changing forest composition and structure across the western US, and suggest that sustained anthropogenic and natural stress will likely result in broad-scale transformation of temperate forests globally.

[1] Department of Forestry, Michigan State University, East Lansing, MI, USA. [2] School of Environmental and Forest Sciences, University of Washington, Seattle, WA, USA. [3] Department of Geography, Environment, and Spatial Sciences, Michigan State University, East Lansing, MI, USA. [4] Forest Service, Northern Research Station, US Department of Agriculture, St Paul, MN, USA. [5] Northeast Temperate Inventory and Monitoring Network, US National Park Service, Woodstock, VT, USA. ✉email: stankehu@uw.edu

Persistent shifts in forest composition, structure, and function depend largely on the demographic response of trees to changing environmental drivers and disturbance regimes[1,2]. Across temperate forests—representing ~25% of the world's forested land area[3]—recent reports of increasing tree mortality have been attributed to complex interactions among climate, native insects and pathogens, and uncharacteristically severe wildfire[4–6]. Such pervasive changes in tree population dynamics can have substantial impacts on the ecosystem services provided by temperate forests, including carbon storage and sequestration[7], climate regulation[8], and provisioning of drinking water[9]. Sustained anthropogenic and natural stress is thus expected to result in broad-scale transformation of temperate forests and the services they provide[10,11]. As such, a key challenge for ecological research is to quantify the patterns and underlying drivers of changing tree populations to better inform forest management and help ease ecological transitions[12].

Tree demographic rates (e.g., mortality) are important, widely used indicators of forest health[12]. However, tree demographic rates are confounded by stand development processes (i.e., stand aging)[13,14] and do not yield a comprehensive depiction of tree population dynamics when considered individually (i.e., the net result of growth, recruitment, and mortality processes; tree abundance shifts)[15–17]. Thus, while recent reports of elevated tree mortality are suggestive of broad-scale changes in the composition and structure of temperate forests[4,5,18], such conclusions should not be accepted in the absence of information regarding tree recruitment, growth, and stand development[1,19]. Previous efforts to quantify trends in tree population dynamics have often relied on observations from old forests to minimize the influence of stand development processes[5,18]. Still, patterns of tree population dynamics observed in old forests are seldom characteristic of those in younger forests[20] and are thus unlikely to be representative of patterns emerging across forest landscapes (or forested regions) that are composed of a mosaic of stands in various stages of development[21]. As such, advancement in detection and prediction of forest health decline depends largely on the dissemination of methods that comprehensively describe tree population dynamics and account for variation in tree demography arising from stand development processes[14,19].

To this end, we propose the forest stability index (FSI), a direct measure of temporal change in relative live tree density that is independent of stand development processes by design. Temporal change in absolute tree density (e.g., trees per hectare (TPH)) emerges from the joint demographic response of tree populations to endogenous (e.g., inter-tree competition) and exogenous drivers (e.g., wildfire)[22]. That is, change in absolute tree density is the net result of mortality, growth, and recruitment processes, and is thus a demographically comprehensive measure of tree population dynamics. Relative tree density may be defined as the proportion of absolute tree density observed in a stand relative to the maximum theoretical density the stand could achieve given its observed tree size-distribution. The maximum density that a population of trees may achieve is expected to decline as individuals grow in size, following well-established allometric scaling laws that drive stand development processes (i.e., self-thinning)[15–17,23,24]. Indices of relative tree density (i.e., ratio of observed and maximum theoretical tree density) are thus independent of stand development processes by definition, as allometric tree size-density relationships are explicitly acknowledged in their denominator.

The FSI is defined as the change in relative density observed in a population of trees over time (e.g., via remeasurement of forest inventory plots). Here stability is achieved when the relative density of a tree population is constant (e.g., FSI equal to zero, stand remains at 50% stocking over time), despite underlying changes in absolute tree density and size distribution. Stability in this sense is uncommon at the stand-scale, as stands progress toward maximum relative density in the absence of exogenous stress (positive FSI, e.g., tree growth exceeds mortality) and relative density is expected to decline given disturbance (negative FSI, e.g., disturbances act as thinning agents). At the landscape-scale, however, stability represents a balance between disturbance and tree growth processes, and deviations from this dynamic equilibrium may be indicative of pervasive changes in forest structure, composition, and function.

We use the FSI to identify patterns in relative tree density shifts of the eight most abundant tree species in the western United States (US) and determine the importance of major forest disturbances in driving the population dynamics of each species over the last two decades. Many forests in the western US have experienced recent increases in the extent, severity, and frequency of wildfire[25,26], drought[27,28], and insect-pest outbreaks[29,30], owing in part to changing climate and past forest management (i.e., fire suppression). Likewise, large-scale tree mortality events[5,31] and recruitment failures[32,33] indicate that widespread forest change is already underway in the region. Such issues are not, however, unique to forests of the western US. Increased disturbance activity has been documented in other regions of the temperate biome in recent decades[12,34,35], and the broad spatial and climatic domain encompassed by the western US suggests that patterns of forest change observed herein may be highly relevant to temperate forests across the globe.

We draw upon over 24,000 repeated censuses of US Forest Service Forest Inventory and Analysis (FIA) plots to address the following questions: (1) What is the current status of populations of the eight most abundant tree species in the western US (i.e., expanding, declining), and what do inter-specific differences in the rate of relative tree density shifts indicate about changes in forest composition? (2) Is the rate of relative tree density shifts size-dependent, and if so, what do these relationships indicate about changes in forest structure? (3) Does the rate of relative tree density shifts vary across space within species ranges, and what are the general patterns of change for each species? (4) How do major forest disturbances influence the populations of these dominant species, and what do these relationships suggest about species sensitivity to future changes in forest disturbance regimes?

Here, we show a majority of the most abundant tree species in the western US experienced significant population decline over the last two decades. Further, we show the magnitude of change in tree populations diverges strongly across species, species-size distributions, and species ranges, and the patterns of such change are generally inconsistent with broad-scale reversion of forests toward historical conditions. Altogether, we provide empirical evidence of widespread, yet spatially varying, changes in forest composition and structure over the last two decades in the western US.

## Results

We identified the eight most abundant tree species in the western US by their estimated total number of live stems (i.e., diameter ≥2.54 cm at 1.37 m above ground) across the region (Fig. 1). Top species represented six distinct genera and three families (Table 1). We categorized species by general ecosystem associations, including two woodland species (i.e., characteristic of mid-high-elevation desert/steppe), three subalpine species (i.e., commonly occurring in cool, moist high-elevation forests), and three montane species (i.e., characteristic of mid-elevation forests climatically bounded by woodland (hotter, drier) and subalpine ecosystems (cooler, wetter)). Together these top eight species

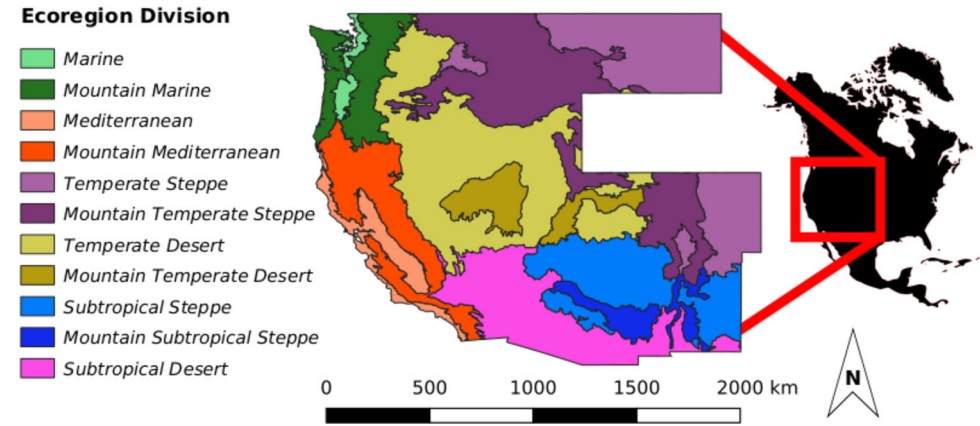

**Fig. 1 Study area colored by ecoregion divisions.** Ecoregion divisions are differentiated by broad-scale patterns of precipitation and temperature. Our study area spans both the humid and dry domains of the western US. We exclude the state of Wyoming due a lack of data.

**Table 1 Scientific and common names of the eight most abundant tree species across the western US, listed in order of decreasing prevalence (i.e., the proportion of total number of stems represented by each species across the region).**

| Common name | Scientific name | Ecosystem association | Prevalence | No. plots |
|---|---|---|---|---|
| Douglas-fir | *Pseudotsuga menziesii* Mirb. | Montane | 0.15 | 12,284 |
| Lodgepole pine | *Pinus contorta* Doug. | Subalpine | 0.11 | 4556 |
| Subalpine fir | *Abies lasiocarpa* Hook. | Subalpine | 0.09 | 3174 |
| Ponderosa pine | *Pinus ponderosa* Doug. | Montane | 0.06 | 7309 |
| Common pinyon | *Pinus edulis* Engelm. | Woodland | 0.05 | 3076 |
| Quaking aspen | *Populus tremuloides* Michx. | Montane | 0.05 | 1723 |
| Engelmann spruce | *Picea Engelmannii* Parry | Subalpine | 0.05 | 3079 |
| Utah juniper | *Juniperus osteosperma* Torr. | Woodland | 0.04 | 3446 |

Commonly accepted ecosystem associations are reported for each species along with their respective sample size in the FIA plot network (remeasured plots only).

accounted for 61.6% of all live trees across the study region (62.3% of total live basal area).

The FSI is defined as the average annual change in the relative density of a population of live trees, or the ratio of observed tree density to maximum potential tree density given site conditions and observed tree size-distributions. Hence, significant positive values of the FSI indicate increased relative density (i.e., population expansion), significant negative values indicate decreased relative density (i.e., population decline), and values of the FSI not significantly different from zero indicate population stability (i.e., no change in relative density). Herein, we treat changes in population range boundaries and within-range density shifts as functionally equivalent processes. For example, range expansion (population occurrence on a site where it was previously absent) is represented by the FSI as a positive change in relative density (i.e., where previous relative density is zero). Thus, when summarized across broad spatial domains the FSI represents a comprehensive measure of the net performance of a population of trees during the temporal frame of sampling (i.e., in terms of net changes in relative abundance).

**Broad-scale shifts in forest composition and structure.** Species-level estimates of the mean FSI across the entire study region (i.e., range-average estimates) reveal broad-scale patterns of rapid change in the composition of western US forests (over 91 million hectares of forestland) over the last two decades (Fig. 2). Three of the eight most abundant species in the region (lodgepole pine, Engelmann spruce, and quaking aspen) exhibited average decreases in relative density (i.e., population decline) at rates exceeding 1% per year over the 18-year study period (2001–2018), whereas Douglas-fir increased in relative density at nearly the

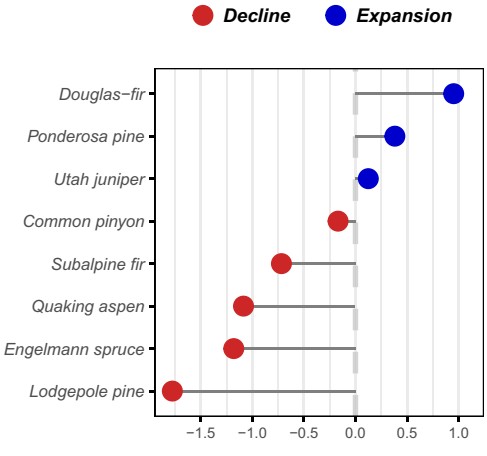

**Fig. 2 Range-average % forest stability index (FSI) of the eight most abundant tree species in the western US over the period 2001–2018.** Population decline (red) occurs when the FSI is negative and the associated 95% confidence interval does not include zero. Conversely, population expansion (blue) occurs when the FSI is positive and the associated confidence interval does not include zero. Here, the %FSI is a direct measure of average annual percent change in the relative density of each species across their ranges in the western US. Thus, total % change in relative density can be estimated by multiplying the %FSI by the length of the study period (18 years). For reference, complete loss of a species over the study period would be indicated by a %FSI value of −5.56%. Source data are provided as a Source Data file.

same rate. For reference, a %FSI equal to 1% is equivalent to an 18% change in relative density over the duration of the study period. Across all species, population decline occurred more frequently (five of eight of species) and with greater magnitude (Fig. 2; leftward skew) than population expansion.

Severe rates of population decline were apparent in all three subalpine species considered herein, with lodgepole pine and Engelmann spruce exhibiting the highest rates of decline among all species (lodgepole pine approaching decline of 2% annually, 36% over the entire study period). Douglas-fir and ponderosa pine, both montane species, exhibited the highest rates of range-wide population expansion. In contrast, quaking aspen populations declined at a rate exceeding the rate of expansion observed in other montane species. Woodland species exhibited the highest degree of population stability (range-average mean FSI values closest to zero) over the last two decades, although low rates (<0.20% annually, 3.6% over the duration of the study period) of population expansion and decline were observed for Utah juniper and common pinyon, respectively.

Variation in range-average estimates of the FSI across species-size distributions are indicative of extensive, complex shifts in the size-structure of forests of the western US over the period 2001–2018 (Fig. 3). Rapid population decline was evident in nearly all size-classes of subalpine tree species (except the smallest 10% of subalpine fir and lodgepole pine). Further, the rate of population decline appeared to increase with increasing tree size across all subalpine species (Fig. 3; downward trend in FSI), and this trend was most severe in lodgepole pine (largest 20% of trees declined at rates exceeding 3% annually, 54% over the duration of the study period). The opposite pattern appeared for Douglas-fir and ponderosa pine, where the largest trees generally out-performed the smallest trees of each species (i.e., higher or more positive changes in relative density).

Interestingly, population decline was evident among the smallest size-classes of Douglas-fir (lower 20%) and ponderosa pine (lower 50%), whereas the largest size-classes of both species exhibited population expansion. Patterns of change in relative density across the size-distribution of quaking aspen appeared to follow an approximately quadratic trend, with population expansion evident in the smallest and largest 10% of stems and severe population decline (approaching 3% annually, 54% over the duration of the study period) apparent near the median (50%) tree size class. The size distribution of woodland species (i.e., common pinyon and Utah juniper) appeared to be the most stable of all species examined, indicated by relatively low variation in relative density shifts across tree size-classes (Fig. 3; nearly flat trends).

**Early indications of shifting species distributions**. Summaries of the FSI within ecoregion divisions revealed broad-scale spatial patterns of change in the relative density of each species over the last two decades in the western US (Fig. 4b). Population decline was spatially pervasive among all subalpine species and particularly severe for lodgepole pine in the mountain temperate steppe division (i.e., central and northern Rocky Mountains). Though interestingly, lodgepole pine populations increased in relative density (i.e., population expansion) in the mountain marine (i.e., coastal Pacific Northwest) and mountain Mediterranean (i.e., Sierra Nevada) divisions over the study period. Quaking aspen exhibited consistent rates of population decline across its range within the study region (~1% annually, 18% over the duration of the study period). Spatial patterns of relative density shifts of Douglas-fir and ponderosa pine were similar, with decline evident for both species in the mountain subtropical steppe division (i.e., southern Rocky Mountains) and expansion in the northwestern portion of the study region (particularly strong for Douglas-fir in

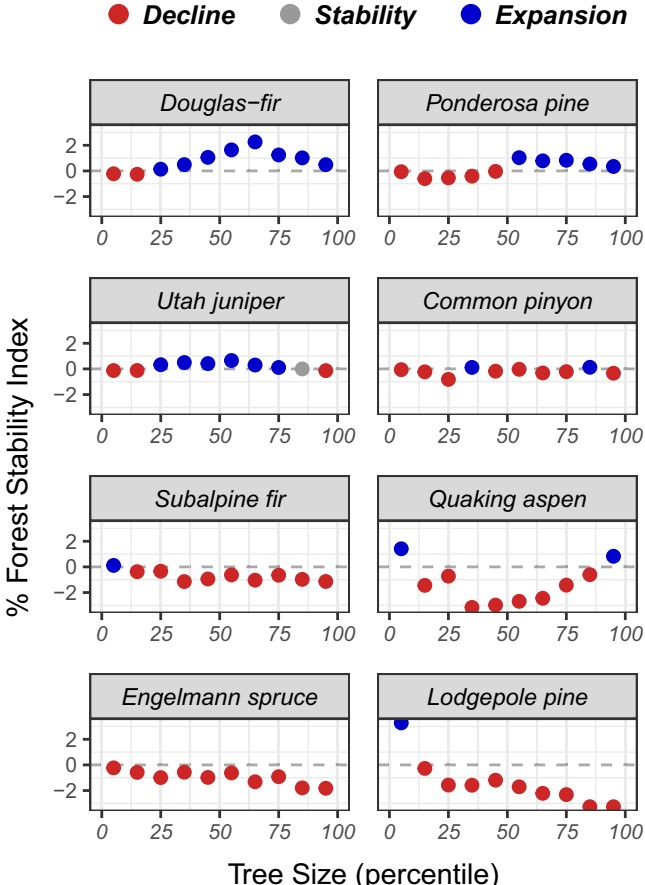

**Fig. 3 Range-average % forest stability index (FSI) across size distributions of the eight most abundant tree species in the western US over the period 2001–2018.** Population decline (red) occurs when the FSI is negative and the associated confidence interval does not include zero. Conversely, population expansion (blue) occurs when the FSI is positive and the associated confidence interval does not include zero. Here, the %FSI is a direct measure of average annual percent change in the relative density of each species across their ranges in the western US, and variation in the FSI across species-size distributions is indicative of shifts in forest structure during the study period. Total % change in relative density can be estimated by multiplying the %FSI by the length of the study period (18 years), with a maximum annual decline of −5.56% (complete loss of the population over the study period). Source data are provided as a Source Data file.

the marine and mountain marine divisions). In contrast, we found general patterns of population stability across the ranges of both woodland species, with the majority of each species' range characterized by FSI values near zero.

While the general spatial patterns of the FSI observed in subsection-level summaries (Fig. 4a) mirrored those of division-level summaries, subsection-level summaries revealed patterns of change in species relative density at finer spatial scales than division-level summaries. Qualitatively, the FSI appears to exhibit strong, positive spatial auto-correlation across ecoregion subsections, and such local-scale variation is not well represented in division-level summaries. Specifically, local regions of population expansion and regions of population decline (i.e., consisting of multiple adjoining ecoregion subsections with similar FSI values) emerge to varying degrees within the range of each species. However, spatial patterns of population performance were not always consistent among overlapping species, indicating shifts in local-scale forest composition.

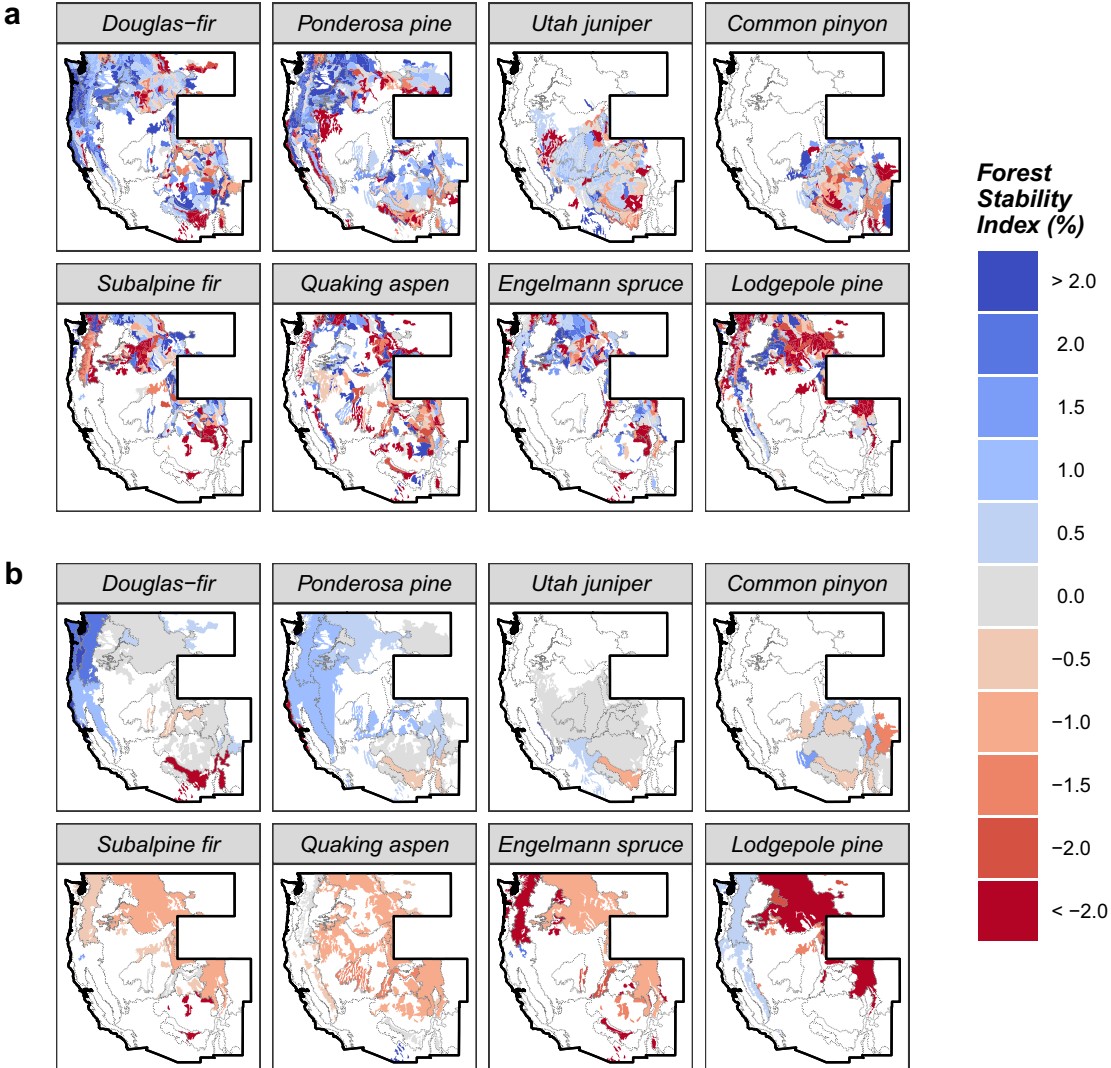

**Fig. 4 Spatial variation in the % forest stability index (%FSI) of eight most abundant species across their ranges in the western US over the period 2001–2018.** Mean %FSI values are mapped at two spatial scales: ecoregion subsection **a** and ecoregion division **b**. Boundaries of the study region are outlined in black, and white areas within the study area indicate absence of a species in the FIA plot network (i.e., colored regions represent species' ranges). For reference, ecoregion division boundaries (as seen in Fig. 1) are outlined as gray dotted lines, and maps of the %FSI within ecoregion divisions (bottom) have been clipped to the extent of each species' range (i.e., defined by ecoregion subsections where the species was detected on an FIA plot). Tree populations have been observed to expand in areas characterized by positive FSI estimates (blue), and decline in areas characterized by negative %FSI estimates (red) during the inventory period (approximately two decades). For reference, %FSI values −2% indicate a 36% decline in relative density over the duration of the study period. Source data are provided as a Source Data file.

**Forest disturbances as drivers of relative tree density shifts.** Estimated effects of forest disturbances on relative density highlight disturbances as important drivers of tree population dynamics in the western US, though the magnitude of disturbance effects varied strongly by tree species and disturbance type (Fig. 5). Here, we present effects of forest disturbances as percent change in relative tree density estimated to be caused by each disturbance type at the population-level (i.e., product of disturbance severity and disturbance probability). Hence, high magnitude of estimated effects indicates a disturbance type is an important driver of the population dynamics of a tree species (e.g., Fig. 5, insect outbreaks in lodgepole pine).

Across all species, fire and insect outbreaks generally emerged as more important drivers (i.e., larger estimated effects) of relative density shifts than disease (Fig. 5), although the effects of disease exceeded those of fire and insects for quaking aspen. In most cases, disturbance was negatively associated with changes in

relative density, however estimated effects of disease were positive in Engelmann spruce and lodgepole pine over the study period. Disturbance effects appeared to be most severe (i.e., highest magnitude, exceeding 0.2% annually) among subalpine species, where insect outbreaks were determined to be dominant drivers of relative density shifts in Engelmann spruce and lodgepole pine, and fire was of heightened importance for subalpine fir. Fire also emerged as the most important disturbance type affecting the relative density of Douglas-fir and ponderosa pine, both montane species. The effects of disturbance appeared to be least severe among Utah juniper and common pinyon, relative to other species (<0.1% annually).

## Discussion
Complex interactions between changing climate, forest disturbance regimes, and past forest management (e.g., fire suppression) are

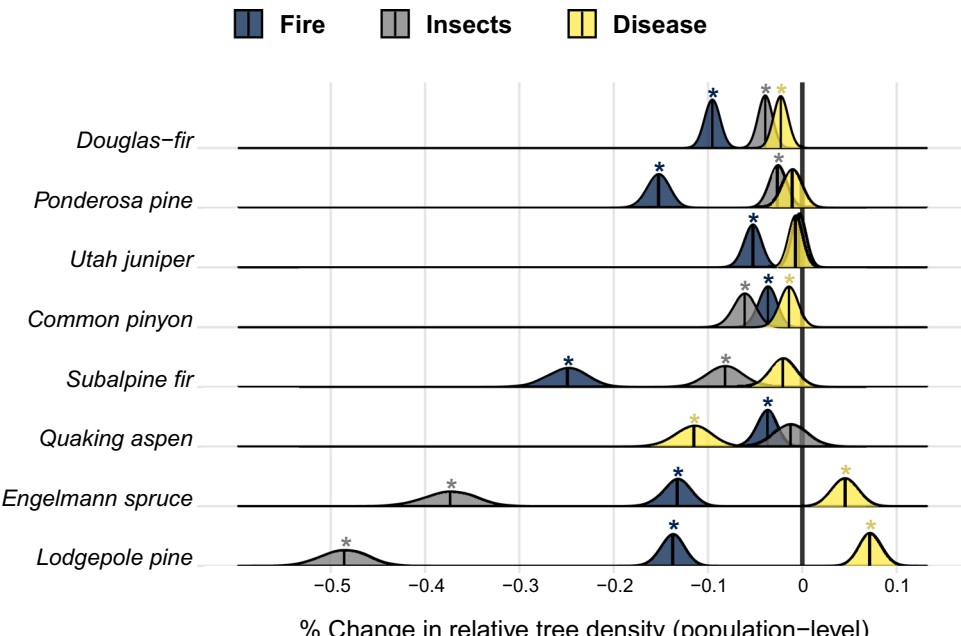

**Fig. 5 Posterior distributions of the estimated effects of forest disturbances on the relative density of live tree populations for the eight most abundant species in the western US over the period 2001–2018.** Change in relative tree density resulting from each disturbance type is estimated as the product of average disturbance severity and probability (annual), where disturbance severity is defined as the average difference in relative density shifts between undisturbed and disturbed sites. Posterior probability distributions of parameters are estimated via Markov chain Monte Carlo (5000 samples). Posterior medians of each parameter are plotted as black vertical lines. Asterisks indicate the 95% credible interval of the mean effect excludes zero, and hence are considered statistically significant. Source data are provided as a Source Data file.

driving accelerated change in tree demography in temperate forests[4,5,36]. However, a great deal of uncertainty remains regarding the net result of such demographic change and its consequences for forest composition and structure across broad spatial domains[1]. Herein, we develop the FSI, a standardized demographic index that weights observed changes in live tree density and tree size against those expected given well-established allometric size-density relationships in forests. We then apply the FSI to over 24,000 remeasured forest inventory plots in the western US to quantify recent trends in tree population dynamics in the region. Our results indicate a majority of the most abundant tree species in the western US experienced significant population decline over the period 2001–2018 (five of eight species, representing 60.7% of all stems of study species), where population decline is indicated by a net decrease in relative live tree density (i.e., negative FSI). Furthermore, we found strong divergence in the magnitude of change in relative tree density across species (Fig. 2), species-size distributions (Fig. 3), and species ranges (Fig. 4). Such dramatic variation in relative density shifts provides empirical evidence of broad-scale, yet spatially varying, changes in forest composition and structure over the last two decades in the western US.

Importantly, the current composition and structure of many forests types in the western US differ drastically from their historic range of variability, arising as a legacy of widespread fire suppression (early 20th century to present day) and intensive harvesting (19th to early 20th century)[37]. Novel forest conditions (e.g., abundance of high density, closed-canopy forests dominated by fire-intolerant species) have interacted with changing climate to incite rapid increases in disturbance activity in the western US and other regions of the temperate biome[29,34]. At face-value, it is therefore not inherently surprising to observe a net decrease in the relative density of many top tree species in the western US, as disturbances act as natural thinning agents in tree populations. However, the examination of patterns in live tree density shifts across species ranges and size-distributions offers a sobering line

of evidence that is inconsistent with broad-scale reversion of forests toward historical conditions. Instead, our results support the following general trends in western US forests over the last two decades: (1) severe, spatially pervasive decline of subalpine forests coinciding with density shifts towards smaller tree size-classes; (2) broad-scale expansion of large-diameter montane conifers and decline of small-diameter conspecifics; and (3) net population stability of woodland species.

The rate of population decline we observed in subalpine species is particularly severe (comprising 25.3% of all trees in the western US). Over the duration of the 18-year study period, the relative density of lodgepole pine declined 32.2% (±0.05%) across its range in the western United States, while Engelmann spruce and subalpine fir declined 21.3% (±0.05%) and 13.2% (±0.04%) across their respective ranges, respectively (Fig. 2). Decline appears spatially pervasive at broad scales across the study region (ecoregion divisions), however local regions of intense population decline and expansion emerge at finer spatial scales (ecoregion subsections) for all subalpine species (Fig. 4). Such patterns may indicate the primary drivers of subalpine tree population dynamics operate at sub-regional scales, likely responding to local-scale variation in climate, topoedaphic conditions, and disturbance history[37]. Previous studies have indicated that subalpine tree species are among the most vulnerable to future changes in climate and forest disturbance regimes[38–40]. Our results indicate this heightened vulnerability is already manifesting across the western US, serving as an early warning of potentially widespread, rapid decline of subalpine forests in other regions of the temperate biome.

Interestingly, we show that severity of population decline increased with tree size for each subalpine species during the study period (Fig. 3). That is, population decline was most severe among the largest trees of each species. This result adds to an increasing body of evidence, suggesting that large, old trees are at high risk of decline in forests across the globe[41]. Large, old trees

generally occur at low density but are of high ecological significance, influencing the rates and patterns of regeneration and succession, moderating microclimate and water use, and contributing disproportionately to forest biomass and carbon cycling at a global scale[42]. Hence, the rapid decline of large-diameter trees we observe in subalpine tree species of the western US is of grave concern and may foreshadow broad-scale transformation in the structure and ecological function of subalpine forests.

In addition, we found population decline to be pervasive across all but the smallest size-classes of subalpine species (Fig. 3). Hence, our results indicate that subalpine forests of the western US have, on average, become younger and thinner over the last two decades. Of all species examined herein, the size-density distributions of subalpine fir and Engelmann spruce are arguably the most likely to exist within their historic range of variability as both species tend to occur in cool, moist forests characterized by infrequent stand-replacing fires (i.e., effects of fire suppression are marginal relative to dry forest)[43]. The pervasive, size-dependent decline we observe in subalpine fir and Engelmann spruce is thus particularly concerning, indicating the size-distribution of each species may be beginning to depart from historically stable conditions. In contrast, decades of fire suppression and intensive harvesting in the western US have resulted in an overabundance of mature, homogeneous lodgepole pine forest that is highly susceptible to native insect outbreaks[37]. As such, reversion to historical conditions would require an increase in heterogeneity in the size-distribution of lodgepole pine at the landscape-level. Instead, the size-dependent patterns of decline we observe in lodgepole pine is indicative of increased homogeneity in the species' size-distribution, where small-diameter stems have become increasingly common relative to large-diameter stems despite a decline in relative tree density across nearly all tree size-classes (younger, thinner, more structurally homogeneous forests).

Our results further indicate that insect outbreaks were >2.5 times more important than other disturbance types in driving relative density shifts of lodgepole pine and Engelmann spruce over the study period (Fig. 5), likely linked to recent outbreaks of mountain pine beetle (*Dendroctonus ponderosae*)[44], and spruce beetle (*Dendroctonus rufipennis*)[45], respectively. As both mountain pine beetle and spruce beetle have shown preference for large hosts (i.e., large-diameter trees), heightened insect-pest activity may explain size-dependent patterns of decline in lodgepole pine and Engelmann spruce. In contrast, we determined wildfire to be more than 3 times more important than other disturbance types in driving relative density shifts of subalpine fir (Fig. 5), and recent increases in the extent and frequency of wildfire[25,46] could potentially explain size-dependent decline observed in the species. Specifically, increases in fire probability (and disturbance probability more generally) are likely to coincide with reduced mean stand age[21] and subsequent decline in populations of large trees across a landscape. Interestingly, we found the relative density of lodgepole pine and Engelmann spruce increase, on average, in response to disease outbreaks, opposite their response to other disturbance types. We argue this result may arise from inter-specific compensatory responses to host-specific pathogens and/or mortality complexes. That is, one species may benefit from the targeted mortality of a competing species within a stand. It is likely that diseases affecting quaking aspen (e.g., sudden aspen decline[47]) and subalpine fir (e.g., subalpine fir decline[48]) may result in a positive growth response of competing lodgepole pine and Engelmann spruce, thereby increasing their relative density within affected stands.

We observed the highest rates of range-average population expansion in Douglas-fir (17.1% ± 0.02 over the study period) and ponderosa pine (6.9% ± 0.03 over the study period; Fig. 2),

widespread montane conifers that together represent 21.2% of all trees across the western US. It is important to note the population expansion observed for Douglas-fir and ponderosa pine may not be desirable in many settings, particularly in dry forests where both species occur frequently as canopy dominants[49,50]. Across the western US, decades of fire exclusion have created overstocked stand conditions that increase the probability of high-severity disturbance (i.e., wildfire, insect outbreak)[51,52] and may degrade forest resilience[37]. In many cases, management aims to reduce tree density via stand thinning and fuels reduction. Hence high rates of relative density increases observed in Douglas-fir and ponderosa pine in the interior Pacific Northwest and portions of the Rocky Mountain region (Fig. 4), may be of substantial concern to forest managers. In contrast, patterns of decreased relative density of ponderosa pine and Douglas-fir observed in the Southern Rocky Mountains may be indicative of tree populations shifting nearer historic relative densities (Fig. 4).

Divergence in both sign and magnitude of relative density shifts across size-distributions of ponderosa pine and Douglas-fir is indicative of broad-scale shifts in the structure of montane coniferous forests in the western US. Specifically, a peak in population expansion is evident among the 50–75th percentile of tree size-classes in both species, and potentially arises as a legacy of widespread logging during the 19th and early 20th century (i.e., owing to simultaneous maturation of stands across a broad spatial domain)[51]. Furthermore, population decline in the smallest size classes of Douglas-fir and ponderosa pine may potentially be linked to heightened wildfire activity during the study period[46], as fire was more than twice as important than other disturbance types in driving change in relative density in Douglas-fir, and more than five times as important in ponderosa pine (Fig. 5). That is, increased wildfire activity may have contributed to desirable change in the structure of montane coniferous forests (i.e., reduced relative density of small trees) over the last two decades in the western US, underscoring the potential for fire (both managed fire and wildfire) to foster forest resilience and contribute to restoration efforts in fire-adapted forests[53].

We found that large-diameter populations of Douglas-fir and ponderosa pine outperformed (i.e., exhibited higher positive change in relative density) small-diameter populations of the same species between 2001 and 2018 in the western US (Fig. 3), opposite the pattern observed in subalpine species. In fact, we observed significant expansion among the largest-diameter populations of both species across the region. This surprising result contradicts previously described patterns of global decline of large-diameter trees[41], suggesting that such patterns may vary strongly across species and ecosystem associations in temperate forests (e.g., subalpine vs. montane ecosystems). Yet, pervasive increases in the relative density of medium and large-diameter montane conifers may be highly undesirable in some settings. Specifically, the structure of many dry, fire prone forest types of the western US have shifted towards dense, closed-canopy stand conditions that are highly susceptible to crown fire and outbreaks of native insect and pathogens[52]. As Douglas-fir and ponderosa pine are common canopy dominants in dry forest types, our results may indicate that such systems have diverged further from their historic natural range of variability over the last two decades.

Elevated rates of population decline were evident for quaking aspen during the study period, pervasive across the range of the species (Fig. 4) and across its size-distribution (Fig. 3) in the western US. It is not inherently surprising to observe population decline in quaking aspen given recent spikes in mortality associated with sudden aspen decline[47,54]. However, the rate of population decline observed herein is particularly severe (20.2% ± 0.04% over the 18-year study period) and serves as a useful baseline to assess the performance of other species. Notably, the

rates of range-average decline observed for Engelmann spruce and lodgepole pine exceed that of quaking aspen. Population decline of quaking aspen has received substantial attention in the forest ecology literature[54–56], however decline of Engelmann spruce has received far less. The disproportionate focus on quaking aspen decline in the literature (and lack of focus on other declining species), emphasizes the need for large-scale, comprehensive assessments of changes in forest composition and structure in the western US and temperate forests elsewhere.

We found woodland species (i.e., Utah juniper and common pinyon) to exhibit the highest degree of population stability across their ranges in the western US, relative to other top species. Pinyon-juniper woodlands of the southwestern US, where both Utah juniper and common pinyon occur as dominant species[50], have experienced dramatic expansions in tree density and equally dramatic tree mortality events over the past century[57,58]. However, inadequate understanding of historic disturbance regimes and tree population dynamics of pinyon-juniper woodlands make it difficult to conclude if such changes are beyond their natural range of variation[57]. It is important to note that our results do not preclude such striking change at spatial and/or temporal domains not addressed herein. Rather, our results indicate the net responses of Utah juniper and common pinyon populations are relatively stable across broad spatial domains (species' ranges) and a relatively short period of time (18 years).

The development of methods to quantify the joint demographic response of tree populations to novel environmental and anthropogenic stressors is among the most important advances required to improve predictions of future change in forest composition, structure, and function, and hence inform the management of forest ecosystem services[14,19,59]. Herein, we present the FSI as one such method. The FSI is a standardized index of temporal change in relative live tree density that can be applied in forests of any community and or structural type. Specifically, the FSI weights observed changes in live tree density (e.g., absolute change in tree abundance) and tree size (e.g., tree basal area) against those expected under allometric size-density laws[16,17]. Hence, although most common indices of forest change (e.g., mortality rate) are confounded by transient dynamics in tree demography arising from metabolic scaling and density-dependent mortality[15,16,20], the FSI is independent of these transient dynamics by design. As such, the FSI may yield simple, accurate measures of shifts in the structure and composition of forests when other common indices of forest change cannot.

The simplicity, flexibility, and highly informative nature of the FSI make it well suited for application in a wide range of ecological settings, and we expect the index to be applied in similar studies to assess broad-scale changes in forest composition and structure in other forested regions across the globe. The FSI relies on temporally replicated data (i.e., ideally from a large number of samples) to derive inference of forest change, however the form of data required by the index are quite simple. For example, all estimates presented herein were derived from basic measures of tree density (e.g., TPH), tree size (e.g., diameter at breast height; DBH), and binary codes indicating the presence or absence of disturbance at measurement sites. As such, the FSI is uniquely well suited for applications using data collected in large-scale national forest inventories. To this end, we provide a flexible implementation of the FSI in the publicly available R package, rFIA[60], for application of the index using data collected by the US Forest Service FIA program.

Increased disturbance activity has been documented across temperate forests in recent decades[12,34,35], and forests of the western US are no exception[29,46]. As disturbances modify forest structure and regulate tree population dynamics[21], our observation

of recent broad-scale shifts in relative tree density across the western US is not inherently surprising. However, the rate and pattern of change we observe across species, species-size distributions, and species ranges is alarming and in many cases undesirable. Furthermore, results of our efforts to quantify the importance of major forest disturbances in driving change in tree populations provide a unique opportunity to assess the vulnerability of tree species to sustained shifts in forest disturbance regimes, as expected under global climate change[2,12]. Importantly, the temporal frame of this study (18 years) is relatively short given the long life-spans of many tree species in the western US (i.e., individuals may live for multiple hundreds of years). As such, it is unclear how the patterns of change we observe herein will translate to long-term trends in forest dynamics in the region, highlighting an important challenge for future research. Nevertheless, our results offer an early warning of recent, widespread change in forest composition and structure across the western US, and suggest that sustained anthropogenic and natural stress are likely to result in broad-scale change of temperate forests globally.

## Methods
**Field observations**. Since 1999, the FIA program has operated an extensive, nationally consistent forest inventory designed to monitor changes in forests across all lands in the US[61]. We used FIA data from 10 states in the continental western US (Washington, Oregon, California, Idaho, Montana, Utah, Nevada, Colorado, Arizona, and New Mexico) to quantify shifts in relative live tree density, excluding Wyoming due to a lack of repeated censuses (Fig. 1). This region spans a wide variety of climatic regimes and forest types, ranging from temperate rain forests of the coastal Pacific Northwest to pinyon-juniper woodlands of the interior southwest[62]. Although the spatial extent of the FIA plot network represents a large portion of the current range of all species examined in this study (Table 1), substantial portions of some species ranges (e.g., Douglas-fir) extend beyond the study region into Canada and/or Mexico and therefore were not fully addressed here.

The FIA program measures forest attributes on a network of permanent ground plots that are systematically distributed at a rate of ~1 plot per 2428 hectares across the US[61]. For trees, 12.7 cm DBH and larger, attributes (e.g., species, DBH, live/dead) are measured on a cluster of four 168 m² subplots[61]. Trees 2.54–12.7 cm DBH are measured on a microplot (13.5 m²) contained within each subplot, and rare events such as very large trees are measured on an optional macroplot (1012 m²) surrounding each subplot[61]. In the event a major disturbance (i.e., >1 acre in size, resulting in mortality or damage to >25% of trees) has occurred between measurements on a plot, FIA field crews record the primary disturbance agent (e.g., fire) and estimated year of the event. In the western US, one-tenth of ground plots are measured each year, with remeasurements first occurring in 2011. Please see Data Availability for more information on forest inventory data accessibility.

**Forest stability index**. Allometric relationships between size and density of live trees make it difficult to interpret many indices of forest change[19]. Live tree density is expected to decline as trees grow in size, owing to increased individual demand for resources and growing space (i.e., competition)[16,23]. The expected magnitude of change in tree density, given some change in average tree size, varies considerably across forest communities[63], site conditions[64], and stand age classes[23]. Thus, we posit it is useful to contextualize observed changes in live tree density relative to those expected given shifts in average tree size within a stand. To this end, we developed the FSI, a measure of change in relative live tree density that can be applied in stands of any forest community and/or structural type.

To compute the FSI, we first develop a model of maximum size-density relationships for tree populations in our study system (Fig. 6). This model describes the theoretical maximum live tree density ($N_{max}$; in terms of tree number per unit area) attainable in stands as a function of their average tree size ($\bar{S}$) and will be used as a reference curve to determine the proportionate live tree density of observed stands (i.e., observed density with respect to theoretical maximum density). We use average tree basal area as an index of tree size (one, however, could also use biomass, volume, or other indices of tree size). For stand-type $i$, the general form of the maximum tree size-density relationship is given by

$$N_{max}(\bar{S}_i) = a_i \cdot \bar{S}_i^{r_i}, \qquad (1)$$

where $a$ is a scaling factor that describes the maximum tree density at $\bar{S} = 1$ and $r$ is a negative exponent controlling the decay in maximum tree density with increasing average tree size. Such allometric size-density relationships (i.e., power functions) are widely accepted as quantitative law describing the behavior of even-aged plant populations under self-thinning conditions[16,17], and have been used extensively to describe relative stand density in forests[23,24]. As detailed below, we allow both $a$ and $r$ to vary with stand-type $i$, as maximum size-density relationships have been

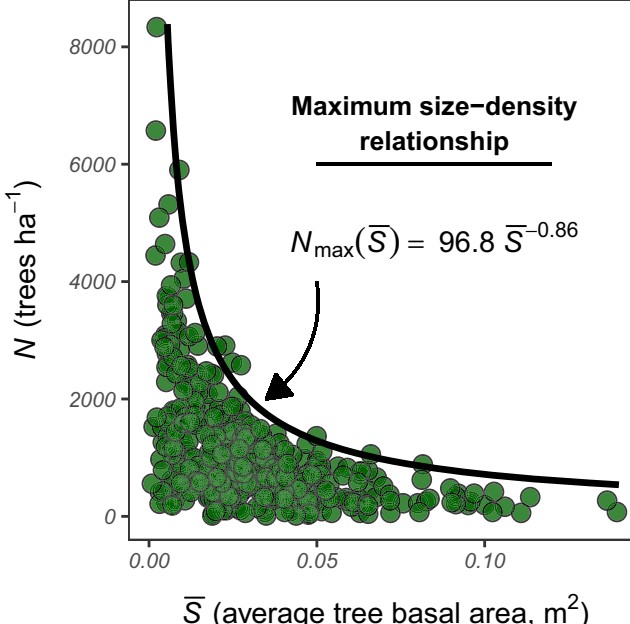

**Fig. 6 Maximum size-density relationship for an example stand-type.**
Individual points represent observed stand-level indices of tree density ($N$)
and average tree size ($\overline{S}$). Maximum tree density ($N_{max}$) is modeled as a
power function of average tree size within a stand. Here, we use quantile
regression to estimate $N_{max}$ as the 99th percentile of $N$ conditional on $\overline{S}$.
The resulting maximum size-density curve can then be used to compute
the relative density of observed stands (RD), where relative density is
defined as ratio of observed tree density ($N$) to maximum theoretical
density ($N_{max}$), given $\overline{S}$. Source data are provided as a Source Data file.

shown to vary across forest communities and ecological settings[63,65]. Allowing $a$
and $r$ to vary by forest community type, for example, allows us to acknowledge that
the maximum tree density attainable in a lodgepole pine stand is likely to differ
from that of a pinyon-juniper stand with the same average tree size.

We next define an index of the relative density of a population of trees $j$ (e.g.,
species, *Pinus edulis*) within a stand of type $i$ (e.g., forest community type, pinyon/
juniper woodland)

$$\text{RD}_{ij} = \sum_{h=1} \frac{N_{hij}}{N_{max}(S_{hi})}, \tag{2}$$

where $N$ is the density represented by tree $h$ (in terms of tree number per unit
area), and $S$ is an index of individual-tree size (e.g., basal area, as used here). The
denominator of Equation (2) represents the maximum tree density attainable in a
stand of type $i$ with average tree size equal to the size of tree $h$. We therefore
express the relative density of a population $j$ within stand-type $i$ as a sum of the
relative densities represented by individual trees within the stand. RD can be
interpreted as the proportionate density, or stocking, of a population of trees within
stand, where values range from 0 (population $j$ is not present within a stand) to 1
(population $j$ constitutes 100% of a stand and the stand is at maximum theoretical
density given its size distribution). As we do in this study, one may apply any range
of estimators to summarize the expected relative density of a population of trees $j$
across a range of different stand-types (e.g., estimate the mean and variance of $RD_j$
across a broad region containing many different stand-types).

It is important to note that Equation (2) is approximately equal to a simpler
method using aggregate indices (i.e., $\frac{\sum_{h=1} N_{hij}}{N_{max}(\overline{S}_i)}$) when tree size-distributions are
normally distributed (even age-structures). However, the use of aggregate indices
introduces class aggregation bias that results in overestimation of relative density in
stands with non-normal size distributions (i.e., uneven age-structures), consistent
with other indices of relative tree density[66]. In contrast, summing tree-level relative
densities eliminates such bias and allows RD to accurately compare
density conditions across stands in very different structural settings (e.g., even-aged
plantation vs. irregularly structured old forest). Furthermore, the partitioning of
relative density into tree-level densities allows RD to be accurately summarized
within tree size-classes[66]. That is, it is possible to explicitly estimate the
contribution of tree size-classes to overall stand density using RD.

For a given population $j$ within stand-type $i$, we define the FSI as the average
annual change in relative tree density observed between successive measurements

of a stand

$$\text{FSI} = \frac{\Delta\text{RD}}{\Delta t}, \tag{3}$$

where $\Delta t$ is the number of years between successive measurement times $t_1$ and $t_2$
and $\Delta\text{RD}$ is the change in RD over $\Delta t$ (i.e., $\text{RD}_{t_2} - \text{RD}_{t_1}$). The FSI may also be
expressed in units of percent change (%FSI), where average annual change in
relative tree density is standardized by previous relative density

$$\%\text{FSI} = \frac{100 \cdot \text{FSI}}{\text{RD}_{t_1}}. \tag{4}$$

Here, stability is defined by zero net change in relative tree density over time (i.e.,
FSI equal to zero), but does not imply zero change in absolute tree density or tree
size distributions. For example, a population exhibiting a decrease in absolute tree
density (e.g., trees per unit area) may be considered stable if such decline is offset
by a compensatory increase in average tree size. However, populations exhibiting
expansion (i.e., $\text{RD}_{t_1} < \text{RD}_{t_2}$) or decline (i.e., $\text{RD}_{t_1} > \text{RD}_{t_2}$) in relative tree density
will be characterized by positive and negative FSI values, respectively.

**Statistical analysis.** We computed the FSI for all remeasured FIA plots in the
western US ($N = 24,229$). We included plots on both public and private lands and
considered all live stems (DBH $\geq 2.54$ cm) in our analysis. As forest management
can effect regional shifts in tree density, we excluded plots with evidence of recent
(i.e., within 5 years of initial measurement) silvicultural treatment (e.g., harvesting,
artificial regeneration, site preparation). All plot measurements occurred from 2001
to 2018, with an average remeasurement interval of 9.78 years (±0.005 years). For
brevity, we restricted our analysis to consider the eight most abundant tree species
in the western US. We identified the most abundant tree species using the rFIA R
package[60], defining abundance in terms of estimated total number of trees (DBH $\geq$
2.54 cm) in the year 2018. We excluded species that exhibit non-tree growth habits
(i.e., shrub, subshrub) across portions of the study region. All statistical analysis
was conducted in Program R (4.0.0)[67].

We developed a Bayesian quantile regression model to estimate maximum size-
density relationships for stand-types observed within our study area. Here, we use
TPH as an index of absolute tree density, average tree basal area ($\overline{\text{BA}}$; equivalent to
tree basal area per hectare divided by TPH) as an index of average tree size, and
forest community type to describe stand-types. We produced stand-level estimates
of TPH and $\overline{\text{BA}}$ from the most recent measurements of FIA plots that (1) lack
evidence of recent (within remeasurement period or preceding 5 years) disturbance
and/or silvicultural treatment and (2) exhibit approximately normal tree diameter
distributions (i.e., even-aged). Here we define an approximately normal tree
diameter distribution as exhibiting Pearson's moment coefficient of skewness
between $-1$ and 1.

We transform the nonlinear size-density relationship to a linear function by
taking the natural logarithm of TPH and $\overline{\text{BA}}$, and use a linear quantile mixed-
effects model to estimate the 99th percentile of TPH conditional on $\overline{\text{BA}}$ (i.e., in log-
log space) for all observed forest community types. We allowed both the model
intercept and coefficient to vary across observed forest community types (i.e.,
random slope/intercept model), thereby acknowledging variation in the scaling
factor ($a$) and exponent ($r$) of the maximum tree size-density relationship across
stand-types. We place informative normal priors on the model intercept ($\mu = 7$,
$\sigma = 1$) and coefficient ($\mu = 0.8025$, $\sigma = 0.1$) following the results of decades of
previous research in maximum tree size-density relationships[16,23,63,65].

The FIA program uses post-stratification to improve precision and reduce non-
response bias in estimates of forest variables[68], and we used these standard post-
stratified estimators to estimate the mean and variance of the FSI for each species
across their respective ranges within the study area (see Code Availability for all
relevant code). Further, the FIA program uses an annual panel system to estimate
current inventories and change, where inventory cycles consist of multiple panels,
and individual panels are comprised of mutually exclusive subsets of ground plots
measured in the same year within a region. Precision of point and change estimates
can often be improved by combining annual panels within an inventory cycle (i.e.,
by augmenting current data with data collected previously). We used the simple
moving average estimator implemented in the rFIA R package[60] to compute
estimates from a series of eight annual panels (i.e., sets of plots remeasured in the
same year) ranging from 2011 to 2018. The simple moving average estimator
combines information from annual panels with equal weight (i.e., irrespective of
time since remeasurement), thereby allowing us to characterize long-term patterns
in relative density shifts. We determine populations to be stable if the 95%
confidence intervals for range-averaged FSI included zero. Alternatively, if
confidence intervals of range-averaged FSI do not include zero, we determine the
population to be expanding when the estimate is positive and declining when the
estimate is negative.

To identify changes in species-size distributions, we used the simple moving
average estimator to estimate the mean and variance of the FSI by species and size
class across the range of each species within our study area. We assign individual
trees to size-classes representing 10% quantiles of observed diameter distributions
(i.e., diameter at 1.37 m above ground) of each species growing on one of seven site
productivity classes (i.e., inherent capacity of a site to grow crops of industrial
wood). That is, we allow size class definitions to vary among species and along a

gradient of site productivity, thereby acknowledging intra-specific variation in diameter distributions arising from differences in growing conditions. The use of quantiles effectively standardizes absolute size distributions, simplifying both intra-specific and inter-specific comparison of trends in relative density shifts along species-size distributions.

We assessed geographic variation in species relative density shifts at two scales: ecoregion divisions and subsections[69]. Ecoregion divisions (shown for our study area in Fig. 1) are large geographic units that represent broad-scale patterns in precipitation and temperature across continents. Ecoregion subsections are subclasses of ecoregion divisions, differentiated by variation in climate, vegetation, terrain, and soils at much finer spatial scales than those represented by divisions. We again used the simple moving average estimator to estimate the mean and variance of the FSI by species within each areal unit (i.e., drawing from FIA plots within each areal unit to estimate mean and variance of the FSI). As a direct measure of changes in relative tree density, spatial variation in the FSI is indicative of spatial shifts in species distributions during the remeasurement interval (i.e., range expansion/contraction and/or within-range relative density shifts). That is, the distribution of populations shift toward regions increasing in relative density and away from regions decreasing in relative density during the temporal frame of sampling. We map estimates of the FSI for each areal unit to assess spatial patterns of changes in relative density and identify regions where widespread geographic shifts in species distributions may be underway.

We sought to quantify the average effect of forest disturbance processes on changes in the relative density of top tree species in the western US over the interval 2001–2018. To this end, we developed a hierarchical Bayesian model to determine the average severity and annual probability of disturbances (i.e., wildfire, insect outbreak, and disease) on sites where each species occurs. Average severity was modeled as

$$y_{jk} \sim \text{normal}(\alpha_j + \sum_l \beta_{jl} \cdot x_{lk}, \; \varsigma_j^2), \tag{5}$$

where $y_{jk}$ is the FSI of species $j$ on plot $k$, $\alpha_j$ is a species-specific intercept, $\beta_{jl}$ is a species-specific coefficient corresponding to the binary variable $x_{lk}$ that takes the value of 1 if disturbance $l$ occurred within plot $k$ measurement interval and 0 otherwise. The intercept and regression coefficients each received an uninformative normal prior distribution. The species-specific residual standard deviation $\varsigma_j$ received a uninformative uniform prior distribution[70].

On average, disturbance will occur at the midpoint of plot remeasurement periods, assuming temporal stationarity in disturbance probability over the study period. As plots in this study are remeasured on 10-year intervals, we assume that tree populations have, on average, 5 years to respond to any disturbance event. Hence, our definition of disturbance severity, $\beta_{jl}$'s, cannot be interpreted as the immediate change in relative tree density resulting from disturbance. Rather, disturbance severity (as defined here) includes the immediate effects of disturbance, as well as 5 years of change in relative tree density prior to and following disturbance (where disturbance is assumed to be functionally instantaneous).

Annual probability of disturbance $l$ on plot $k$ was modeled as

$$x_{lk} \sim \text{binomial}(\Delta t_k, \psi_{jl}), \tag{6}$$

where $\Delta t_k$ is the number of years between successive measurements of plot $k$, viewed here as the number of binomial "trials," and $\psi_{jl}$ is the species-specific probability for disturbance which was assigned a beta(1,1) prior distribution. Hence, annual probability of disturbance is assumed to vary by species $j$ and by disturbance type $l$.

We estimate the mean effect of forest disturbance processes on changes in species-specific relative tree density by multiplying the posterior distributions of $\beta_{jl}$ and $\psi_{jl}$. That is, we multiply species-specific disturbance severity by disturbance probability to yield an estimate of the mean change in relative density caused by disturbance over the study period. We then standardize these values across species by dividing by the average relative density of each species at the beginning of the study period. Thus, standardized values can be interpreted as the annual proportionate change in the relative tree density of each species resulting from disturbance over the period 2001–2018.

**Reporting summary**. Further information on research design is available in the Nature Research Reporting Summary linked to this article.

## Data availability
All forest inventory data used herein are publicly available in comma-delimited text format via the USDA Forest Service FIA data repository (https://apps.fs.usda.gov/fia/datamart/CSV/datamart_csv.html). Source data are provided with this paper.

## Code availability
We have implemented computational routines for the FSI in the publicly available R package, rFIA[60]. In addition, all custom code used herein is publicly available in the following permanent GitHub repository: https://github.com/hunter-stanke/Code-repository---NCOMMS-20-20430.

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

## Acknowledgements
This work was supported by: National Science Foundation grants DMS-1916395, EF-1253225, EF-1241874; US Department of Agriculture, Forest Service, Region 9, Forest Health Protection, Northern Research Station; US National Park Service; and Michigan State University AgBioResearch. The findings and conclusions in thispublication are those of the author(s) and should not be construed to represent any official US Department of Agriculture or U.S. Government determination or policy.

## Author contributions
H.S., A.O.F., G.M.D., A.S.W., and D.W.M. designed research; H.S. performed research; H.S. analyzed data; H.S., A.O.F., G.M.D., A.S.W., and D.W.M. wrote and reviewed the manuscript.

## Competing interests
The authors declare no competing interests.
