## [Peer Review File · Nature Communications]

REVIEWER COMMENTS

Reviewer #1 (Remarks to the Author):

Review Nature Communication – NCOMMS-20_20430

Over half of most abundant trees species in decline in the western United States

The study by Stanke et al. aims to use a new demographic index for the western part of the USA to show how the populations of eight most common tree species have developed over the last two decades. With the data, they create the possibility to examine to what extent individual tree species have increased or decreased in abundance and to what extent regional shifts in the spatial distribution of tree species have occurred. The Forest Stability Index (FSI) chosen for this purpose is a new approach - at least I don't know of any publication that mentions such an index, and even a targeted search has led to the fact that I have not found such a publication. In my opinion, the paper is of special interest to others because it is a simple but very meaningful index applied to a very comprehensive set of data, the application of which will be publicly available (including the R package), and it provides a very good basis for an early warning system regarding the demographic development of tree species on a large spatial scale.

I expect that the paper will be inspiring to apply the same or a slightly modified index in other countries and to interpret the results against the background of changing and still changing environmental conditions. Due to its "simplicity", but nevertheless high informative value (very clearly illustrated in Figure 6), I also assume that this procedure will have a very high acceptance in practice-relevant areas. The claim formulated by the authors is undoubtedly fulfilled and the results will be discussed in an appropriate manner, taking into account the relevant literature to date. I do not see the need for further experiments, but I do have an essential remark concerning the evaluation of the data, especially when using the Linear Mixed Model (see remarks below).

The manuscript is written very clearly, I have only a few critical remarks to make in this respect (see below) and a shortening of the text is not necessary in my view. The objectives formulated by the authors have been met and they have given sufficient consideration to studies that have already been published on the topic. They have also provided all necessary details on the methodological approach, in particular by making the data publicly available and even offering an R package to reproduce the results produced. The statistical analyses are well-founded, but there is a question from my side (see remarks/questions below).

General remarks or questions

(i) The title of the study is explicitly aimed at the western part of the USA - ok. However, the contents are certainly significant for other regions, especially the temperate climate zone. This is also made clear by the authors in the first sentence of the abstract. Against the background that forests of the temperate zone account for about 20% of all forests (based on 3.45 billion ha of forest worldwide according to FAO), the introduction should start in a short form with a global view and give some "key data". Furthermore, the first sentence - as it is currently formulated - should make clear whether the statement "worldwide" is valid or only for the western USA, which at least reference no. 2 suggests. Also with regard to the conclusions or prospects, I think it is more attractive for other readers if these are in principle aimed at a larger part of the temperate zone (but only if this is possible).

(ii) It is indeed a pity and also a certain shortcoming that there is no data/information on management, as it is a very important factor in the composition of tree species. My question about this, since both forests on public and private land have been investigated: Does the management differ in terms of land ownership and would a comparison of these data pools be a way to estimate management effects?

(iii) The results for the two spatial scales are very meaningful in Figures 2 and 3. In particular, however, the elaborate presentation in Figure 3 on the Ecoregion subsections contradicts to a large extent the mainly descriptive statements on p. 7 (last paragraph) in the discussion. I see the finer spatial scale as a great advantage with regard to possible explanations, since one could gain good insights into the varying behaviour of tree species from different initial site conditions. Within a subsection there are local differences, e.g. drier or moist locations due to topography and soil conditions. It is conceivable that locally drier locations anyway have made/make the trees much more susceptible to a decrease due to climate change than those on moist locations. In this context, it is not really clear to me why the LMM (referring to Figures 4 and 5) did not test for interactions between severity of disturbance and long-term climate patterns - at least the text (p.3 below to p. 4) does not mention anything about this, but the aspect of possible interactions between drought and susceptibility to e.g. insect infestation is mentioned as a discussion point from time to time.

And in this context, it is not clear to me why the relationship between disturbance and climate cannot be applied to the small-scale ecoregion subsections, when climate data are available in 4 km resolution. Are there also relatively high-resolution spatial data on the degree of disturbance? I know from my own work how costly such a thing can be, but I still ask, because one of the benefits of the study is the creation of an early warning system - but what good is that if you don't know more concretely what you can do? If the abundance of tree species declines due to drought in the context of global climate change, then it will not be possible to change this situation locally so quickly. However, if it is known that individual species can or cannot replace other species through very good natural regeneration, then action should either be "based on natural regeneration" or targeted planting should be carried out to cushion the consequences of the decline of other tree species.

(iv) When I read in the legend to Figure 3 that "white areas" ... reflect "indicate lack of data", but on page 7 in the 6th to 7th line from below it says "We also observed ... occurred (all species except lodgepole pine), then this is irritating. It makes a difference whether a species is in a subsection (for arealgeographical/ecological reasons) or just no data is available. This should be clearly stated.

Specific remarks

p. 2, 3rd paragraph, 1st sentence; ... the 10 most abundant ..." – should be eight – see also first question in the 4th paragraph

p. 8, 1st paragraph, last sentence; is there one of the references in your list that could be cited here to support the statement? E.g. reference 5, 6 or 49?

p. 9, Climate data, 1st sentence; what time period you choose for the 30-year normal?

p. 14, no. 46.; pinon-juniper, but see no. 47. piñon-juniper

Reviewer #2 (Remarks to the Author):

The manuscript by Stanke et al. is an interesting analysis of changes in species abundance for dominant conifers across the western US. I found the method development to be the most compelling part of this research. The FSI index represents a standardized means for looking at changes in the density and dominance of species within the FIA plot network.

That said, I have serious reservations about the interpretation of the results, the novelty of these findings, and in particular the framing used to convey the findings. At the most basic level, the authors are stating that populations of some of the most dominant species in the western US are in decline due to disturbance. This is not surprising or a novel finding in my view. To support this finding they note that the FSI index calculated from 2001-2018 is negative for many of the conifer species in question. I don't dispute this. My criticism lies in the framing of the problem. The authors tend to equate 'population performance' with abundance. An extensive body of research in western forests has noted that active fire suppression since the 1940s has resulted in current forests with substantially higher densities than the pre-suppression era. Moreover, today's forests have species compositions that are notably different than historical conditions and are often dominated by shade tolerant species at higher proportions than historic conditions. Thus the recent decline in shade tolerant species these authors note needs to be contextualized with respect to historical conditions. Moreover, the period from 1940-1980 was a relatively cool wet period for much of the study domain. The study interval here (2001-2018) represent a period of recent rapid warming and

changes in the fire regime of these forests after this period of active fire suppression with mesic conditions. One could argue that a reduction in tree density is actually returning forest density and composition to conditions that were more common in the past which could be considered simply a reversion to historical norms. This study does not provide a means to contextualize the changes in abundance presented and the short time interval of this study (18 years) doesn't provide the context needed to interpret the changes observed. Moreover, the formulation of the FSI index combines both changes in density (TPH) and basal area. I would argue that the loss of large trees is a bigger issue with respect to ecological functioning in these systems compared to changes in TPH per se. There is clear evidence that we are losing large trees in our forests and this has a number of negative implications for the resilience of these systems. The FSI index doesn't allow us to isolate this effect from changes in density. I would find an analysis that focused on large trees to be much more compelling than the one presented here. In sum the framing of this study seems to suggest that greater abundance of trees = greater population performance. This framing may work in eastern forests but the active disturbance regimes in western forests makes abundance a relatively weak metric of forest health and resilience especially when viewed through a short interval of time.

Specific Comments (The manuscript did not have line numbers so my specific comments will refer to pages and paragraphs)

1) Intro paragraph 2 - I appreciate the general notion that we need to understand multiple facets of vital rates to understand population dynamics but what compensatory mechanisms can offset declines in recruitment rates? One may be higher longevity of trees but all evidence as you note seems to point against that including recent work that suggests larger trees are preferentially being lost in forests globally.

2) Page 2 paragraph 2 – “For example, elevated population performance in the cooler portions of a species' range relative to warmer portions may indicate species migration along the temperature gradient over time (i.e., shift in species abundance distribution toward cooler regions).” You level criticisms against space-for-time substitutions in the previous paragraph but here you are basically advocating for just that.

3) Second to last paragraph in the intro. In the text here you state the 10 most abundant species but your results are for the 8 most abundant.

4) Figure 2 caption. I would suggest providing a plain language description of what the FSI index means and how to interpret the values shown.

5) Figure 4 caption and throughout text. This term “range wide” is misleading as it implies declines across the entire range. Figure 3 shows that there is clear spatial variability in FSI with opposing signs of the index. Is there a better way to describe this? Maybe 'range averaged'

6) I find it interesting that Douglas Fir and Ponderosa pine are the two dominant conifers that experienced notable increases in abundance. Doug fir is an extremely plastic species found over an incredible range of environmental conditions. It is also shade tolerant. Im not surprised by its increase in abundance. Ponderosa pine however is surprising given that there are many studies that show it is struggling to naturally regenerate at the hotter drier sites in which it is found. Did you exclude FIA plots with plantings? These two species are the most widely planted post-fire.

7) Figure 5 suggests that Douglas Fir, lodgepole pine, and subalpine fir are all showing improved performance (higher abundance) in hotter and drier areas within their range. As you note this is counter to expectations and other studies that show increased mortality and reduced recruitment rates for these species in hotter drier environments. How do you reconcile your results with this broader literature?

8) Page 7 last paragraph. Climate is intrinsically local and the gradients experienced over short distances along an elevational transect are dramatic compared to those experienced over large latitudinal gradients. Is there any elevational patterning in the FSI index?

9) Page 8 paragraph 1. The term “heightened severity” implies that rates are different than previous time intervals...but we dont know that from this study and no citations to supporting work are provided.

10) I don't find the section describing how FSI varies with climate compelling. You find a pattern that is counter to a broad array of literature but provide little in the way of why this is the case. My suspicion is that the patterns described with respect to climate are being conflated with the legacy effects of fire suppression and indirect effects of climate on disturbance dynamics. This needs more attention.

11) “The FSI is a measure of population performance”. These are loaded terms as I noted at the beginning of this review. Why not call the FSI a measure of species abundance as this is a clearer depiction of what it is.

12) Methods. You standardize changes in tree density and BA using the stdev of these values across all plots as measured over the study interval. Does this imply that changes in these variables are stationary processes? As I noted, long term exogenous processes like changing disturbance regimes and changing climate will strain a stationarity assumption.

Reviewer #3 (Remarks to the Author):

The study “Over half of most abundant tree species in decline in the western United States” by Stanke et al. presents an analysis of recent demographic changes in the forests of the western US. The authors tackle a highly relevant topic, given forest disturbance regimes are increasing around the globe and have the potential to threaten the integrity and resilience of forest ecosystems. The study is unique in that it relies on a very large observational dataset of repeated measurements, i.e. it reports on actual changes (rather than using the much more widely used space-for-time substitution). Specifically, the authors develop a new index of demographic change in forests, and apply it to the eight most important species of the Western US. They show that a large number of these species are declining. The authors attribute these changes to forest disturbances, mainly wildfire and insect outbreaks. As mentioned the study is highly relevant; it has, however, a number of grave shortcomings in its current form.

First, and most importantly, I find that the key conclusions (regarding the main trend in demographics and their key drivers) are premeditated by the data and analysis performed. The time frame of change analyzed here is 9.78 years (i.e., the mean time span between two inventory censuses), and the data were collected between 2001 and 2018. The authors present a clever new index of demographic change that balances changes in stem density with changes in basal area (more comments on the index below). While I like the overall idea of the index, it can only give negative values (indicating decline) for a period of high disturbance activity (such as 2001 – 2018, which we know from other analyses) and short observational intervals (<10 years), as it takes a forest longer than 10 years to compensate disturbance-induced relative basal area losses through increasing relative stem number via regeneration. If you would be able to allow for a longer regeneration time window, such as 30 years, things might actually become interesting, but this is not yet possible with the currently available data. For the same reason, the change attribution part of the paper is not really meaningful because the index *has* to give negative values with disturbance. Identifying disturbance as the most important driver is thus circular reasoning.

Second, while I generally like the idea of the index and the trade-offs between standardized change in stem number and standardized change in basal area, I find the index to be defined ad hoc, with the authors never actually showing that the underlying premises hold true. In other words: Is demographic constancy really defined by a 1:1 trade-off between standardized basal area change and standardized stem density change? I would think not, because stem number decreases roughly with the 1.605 power of diameter (Reinekes rule) while basal areas increases with the 2.000 power of diameter. So rather than just allude to scaling laws throughout the text I suggest to actually base the index on them, in order to justify the ad hoc assumptions that go into it.

Third, while the authors maintain that their analysis factors out regular transient changes in forest ecosystems, such as succession, I find that I strongly disagree with this notion. The index is indeed clever, but when applied at the level of individual species, such as done here, it does not consider regular successional patterns in forests. When an old-growth system is burned, for instance, it will necessarily lead to a decline in the dominant late-seral species, and no one would expect these species to again gain dominance within 10 years of a disturbance. Rather, early-seral species will take over for some time (decades to centuries), and only gradually the late-seral species that dominated previously will reinvade the area and attain dominance once more. In your analysis this type of succession (which happens regularly in most forests around the world) would show up as a steep decline in the late-seral species, while it is only a temporal decline that prevails only as long as the early seral stages are maintained.

In addition to these methodological flaws, the text is vague in many respects. It, for instance, completely misses a temporal reference throughout; only in the Methods the time frame of analysis is mentioned. Furthermore, the text talks about population increase and decline, however it is not clear for the reader whether a decline means a reduction in the range of occurrence or a reduction in the population level – two very different things. These aspects should be made clear early in the text in order to avoid confusion for the reader.

Lastly, the study is distinctly local in its framing and contextualization. Despite the fact that the last sentence of the abstract alludes to the global temperate forest, the narrative and references used in the introduction and discussion are purely centered on the specific discussion in the Western US. This might be appropriate for a society journal such as an outlet from the Ecological Society of America, but not for a globally relevant journal from the Nature publishing group. Why would this be relevant for a reader in Asia or Australia, for instance?

One last note before I list a number of more detailed comments on the text: Next time you submit a manuscript, please provide line numbers so that comments can be made in reference to specific lines!

p1: individual-tree demographic responses... unclear, as far as I can see demographic responses are at the level of a population, but not at the level of an individual tree

p2: importance of endogenous processes during maturation... I very much agree here – one important process that is grossly neglected is succession, i.e. any reasonable forest ecologist would not expect the same species to dominate in an early-seral patch vs. an old-growth patch! Your analysis also does not account for this process – see my general comment #3 above.

p2: differ among the 10 most abundant tree species in the western US... in the results section it seems that you only focus on the 8 most abundant species

p2: estimated total number of live stems... I don't think that selecting the most important ones purely based on stem density is meaningful. The reason is that this will strongly bias towards early seral species (generally high densities) over late-seral species (generally low densities). I would suggest that you need to at least also include some measure of dimension, e.g., basal area would be an integrated measure over density and size. As it is, the measure of importance (purely based on stem density) is not consistent with the stability index calculated (based on stem density *and* basal area).

p3: Tendency towards range-wide population decline occurred more frequently.... you talk about decline and expansion here, but it is not clear for the reader at this point what your temporal reference is; are we talking about the last five years, the last 50 years, the last 500 years here? Would be important to know to interpret your results!

Figure 2: You talk about confidence intervals in the figure caption, but I can't see them in the figure.

Figure 2: Give an example in the figure caption what a FSI of -0.2 means. As it is the numbers are very hard to contextualize for the reader.

p3: a negative relationship between the FSI and forest disturbance severity for all species and disturbance types... this is not a result, this is a foregone conclusion based on how you defined your FSI. From what I can see it is technically not possible that a disturbance results in stable/ positive FSI responses. So the analysis is circular. See also my general comment above.

p3: fire emerged as the most important predictor for all species except lodgepole pine... again, these statements lack a temporal reference. Fire when? In the year prior to the last observation? 20 years ago? 200 years ago? These forests are fire-driven systems, so the fact that fire resets forest dynamics and instigates regeneration is a part of their natural dynamics.

p4: in historically warmer portions of the species' ranges... what is historically warmer portions, which part of history are you referring to? Last year? The Holocene? And are there really portions of your study region that were warmer than they are currently in the last, say, 20 years?

Figure 3: ecoregion subsection (top) and ecoregion division... I find the two aggregation levels are not needed – they show the same information, only at different grain. Go for the finer grain (and put the coarser grain into the supplement, if needed).

Figure 3: It is unclear how you aggregated the inventory plots to ecoregions. You talk about moving window aggregation in the methods, but I don't understand how you get from moving window values to ecoregions.

Figure 3, caption: here and throughout the text it is not clear if decline and expand means in area, or in terms of demographic indicators (e.g., fewer stems). These are two different signals, and it should be clear what is reported here.

p4: Our results indicate a majority (5 of 8) of the most abundant tree species in the western US are undergoing significant range-wide population decline... I suggest to not base your conclusions on majority votes, but on something like % area experiencing decline/ expansion. This is much more robust/ relevant.

p7: and is independent of transient dynamics associated with stand structural development... how so? I cannot see how the analysis would not reflect transient successional dynamics of a forest. See my general comment above.

p8: Among the primary values of the FSI is its independence of transient dynamics associated with stand structural development and succession... this is a strong claim that you make repeatedly throughout the text, but I cannot see how this claim can be substantiated. If possible, please demonstrate that the index is independent of succession, e.g. based on an observed (or simulated) successional trajectory that is otherwise stable over time.

p10: denotes the standard deviation of the variable x across all plots sampled... sd between plots (in space) or sd over time per plot, or both? This matters a lot for the interpretation of your index, and is unclear.

Statement on the Review of
⟨NCOMMS-20-20430⟩
based on Referee's Report

Hunter Stanke, Andrew O. Finley, Grant M. Domke, Aaron S. Weed, David W. Macfarlane

September 17, 2020

This statement concerns our revision of the ⟨NCOMMS-20-20430⟩ paper, entitled “*Over half of most abundant tree species in decline in the western United States,*” based on the referees' report. All reviewer comments have been listed in their original form (black text), along with individual responses to each comment (blue text). In all responses that follow, line numbers refer to those in the revised manuscript where appropriate modifications have been made.

In light of reviewer comments, we have made major revisions to all sections of the manuscript and offer a complete re-analysis. In addition to addressing specific reviewer comments (below), we offer the following major changes to the manuscript:

1. We developed an improved implementation of the Forest Stability Index (FSI). The new version of the FSI is a direct measure of temporal change in the relative density of live trees, and is based in decades of research on allometric size-density laws in tree populations. Please refer to lines 323-363 for an updated description of the FSI.
2. Per comments from Reviewer 2, we have included an additional analysis in our revised manuscript that addresses change in species size-distributions across their ranges in the western US over the period 2001-2018. Please see lines 401-408 in the revised manuscript for a methodological description and lines 103-116 for results specific to this analysis.
3. Per comments from all reviewers, we have removed our analysis of trends in the FSI along climatic gradients. Per comments and suggestions from Reviewer 3, we have substantially revised our analysis of disturbance effects on the FSI. Please see lines 420-438 in the revised manuscript for a methodological description and lines 131-151 for results specific to this analysis.
4. Per comments from all reviewers, we have re-framed our entire manuscript to improve temporal specificity, provide context of historical conditions in the western US, and support the relevance of our results in temperate forests globally.

Comments by Reviewer #1

General: The study by Stanke et al. aims to use a new demographic index for the western part of the USA to show how the populations of eight most common tree species have developed over the last two decades. With the data, they create the possibility to examine to what extent individual tree species have increased or decreased in abundance and to what extent regional shifts in the spatial distribution of tree species have occurred. The Forest Stability Index (FSI) chosen for this purpose is a new approach - at least I don't know of any publication that mentions such an index, and even a targeted search has led to the fact that I have not found such a publication. In my opinion, the paper is of special interest to others because it is a simple but very meaningful index applied to a very comprehensive set of data, the application of which will be publicly available (including the R package), and it provides a very good basis for an early warning system regarding the demographic development of tree species on a large spatial scale.

I expect that the paper will be inspiring to apply the same or a slightly modified index in other countries and to interpret the results against the background of changing and still changing environmental conditions. Due to its "simplicity", but nevertheless high informative value (very clearly illustrated in Figure 6), I also assume that this procedure will have a very high acceptance in practice-relevant areas. The claim formulated by the authors is undoubtedly fulfilled and the results will be discussed in an appropriate manner, taking into account the relevant literature to date. I do not see the need for further experiments, but I do have an essential remark concerning the evaluation of the data, especially when using the Linear Mixed Model (see remarks below).

The manuscript is written very clearly, I have only a few critical remarks to make in this respect (see below) and a shortening of the text is not necessary in my view. The objectives formulated by the authors have been met and they have given sufficient consideration to studies that have already been published on the topic. They have also provided all necessary details on the methodological approach, in particular by making the data publicly available and even offering an R package to reproduce the results produced. The statistical analyses are well-founded, but there is a question from my side (see remarks/questions below).

Thank you for your kind words and thorough review of our work. We very much appreciate your time and efforts in reviewing this manuscript. In light of all reviewers comments and suggestions, we have provided an improved implementation of the Forest Stability Index, modified the scope of our analysis (plus associated complete re-analysis), and made major revisions to all sections of our manuscript. Each critique noted above is addressed in detail in the comments that follow.

Title: The title of the study is explicitly aimed at the western part of the USA - ok. However, the contents are certainly significant for other regions, especially the temperate climate zone. This is also made clear by the authors in the first sentence of the abstract. Against the background that forests of the temperate zone account for about 20% of all forests (based on 3.45 billion ha of forest worldwide according to FAO), the introduction should start in a short form with a global view and give some "key data". Furthermore, the first sentence - as it is currently formulated - should make clear whether the statement "worldwide" is valid or only for the western USA, which at least reference no. 2 suggests. Also with regard to the conclusions or prospects, I think it is more attractive for other readers if these are in principle aimed at a larger part of the temperate zone (but only if this is possible).

Thank you, we certainly agree the results of this work are significant for other regions of the temperate biome. We have re-written our introduction, discussion, and conclusion to make this clear. Specifically, we broaden our references to provide support for statements across temperate forests. Per your suggestion, we provide key data to demonstrate the importance of temperate forests on a global scale and attempt to explicitly mention the relevance of our work across temperate forests. Please see lines 14-17 of the revised manuscript for a specific example.

It is indeed a pity and also a certain shortcoming that there is no data/information on management, as it is a very important factor in the composition of tree species. My question about this, since both forests on public and private land have been investigated: Does the management differ in terms of land ownership and would a comparison of these data pools be a way to estimate management effects?

We appreciate this comment and agree that the lack of information on management was a considerable shortcoming of the originally submitted manuscript. While comparing estimates of the FSI across land ownership types may be interesting in its own right, we do not believe this is an appropriate method to estimate management effects as forest management occurs to varying degrees on both public and private lands in the western US. Alternatively, we have opted to restrict our analysis to FIA plots with no evidence of forest management (i.e., silvicultural intervention) within the plot remeasurement interval or preceding 5 years to reduce the effects of management in our results. Please see lines 366-368 for further description of these changes in the revised manuscript.

The results for the two spatial scales are very meaningful in Figures 2 and 3. In particular, however, the elaborate presentation in Figure 3 on the Ecoregion subsections contradicts to a large extent the mainly descriptive statements on p. 7 (last paragraph) in the discussion. I see the finer spatial scale as a great advantage with regard to possible explanations, since one could gain good insights into the varying behaviour of tree species from different initial site conditions. Within a subsection there are local differences, e.g. drier or moist locations due to topography and soil conditions. It is conceivable that locally drier locations anyway have made/make the trees much more susceptible to a decrease due to climate change than those on moist locations. In this context, it is not really clear to me why the LMM (referring to Figures 4 and 5) did not test for interactions between severity of disturbance and long-term climate patterns - at least the text (p.3 below top. 4) does not mention anything about this, but the aspect of possible interactions between drought and susceptibility to e.g. insect infestation is mentioned as a discussion point from time to time. And in this context, it is not clear to me why the relationship between disturbance and climate cannot be applied to the small-scale ecoregion subsections, when climate data are available in 4 km resolution. Are there also relatively high-resolution spatial data on the degree of disturbance? I know from my own work how costly such a thing can be, but I still ask, because one of the benefits of the study is the creation of an early warning system - but what good is that if you don't know more concretely what you can do? If the abundance of tree species declines due to drought in the context of global climate change, then it will not be possible to change this situation locally so quickly. However, if it is known that individual species can or cannot replace other species through very good natural regeneration, then action should either be "based on natural regeneration" or targeted planting should be carried out to cushion the consequences of the decline of other tree species.

Thank you, we appreciate these comments. First, we have modified the discussion in the revised version of the manuscript to place less emphasis on shifting species distributions. In doing so, we have removed the mainly descriptive statements previously listed on p. 7 (last paragraph) that you refer to here. In the originally submitted version of the manuscript, we tested for multicollinearity among disturbance and climate variables using variance inflation factors, however in the revised version of the manuscript we have removed our analysis of climate effects on species abundance shifts (per reviewer suggestions). Hence, it is no longer necessary to test for such interactions in our model.

Second, we certainly agree that it would be valuable to adapt our analysis to examine the space-varying relationships of disturbance, climate, and species abundance shifts. However, the development of such a model would be a very substantial undertaking given the immense amount of data used in this study and computational requirements of appropriate statistical methods (e.g., geostatistical model or Markov random field model able to estimate spatial association among plots or regions, respectively). While we see the value in such analysis we feel it should be pursued in a subsequent study due to its substantial scope. Further, we do not feel it necessary to include such an analysis in this study, as most inferences we draw are at the regional-level. Specifically, in the revised version of the manuscript, we primarily use the results of our model relating the FSI to disturbance activity to explain patterns in relative density shifts observed across species ranges within the western US (e.g., Figures 2-3).

When I read in the legend to Figure 3 that "white areas" ... reflect "indicate lack of data", but on page 7 in the 6th to 7th line from below it says "We also observed ... occurred (all species except lodgepole pine), then this is irritating. It makes a difference whether a species is in a subsection (for areal geographical/ecological reasons) or just no data is available. This should be clearly stated.

Thank you for this comment. All “white areas” in Figure 4 (previously Figure 3) indicate that a species was not detected in the FIA plot network in that ecoregion subsection, and hence is assumed to be absent in the region. We have updated the caption of Figure 4 to clarify this.

p. 2, 3rd paragraph, 1st sentence; ... the 10 most abundant ...” - should be eight - see also first question in the 4th paragraph

Excellent catch, thank you. We have updated the text to the “8 most abundant”. Please see line 53 in the revised manuscript.

p. 8, 1st paragraph, last sentence; is there one of the references in your list that could be cited here to support the statement? E.g. reference 5, 6 or 49?

Thank you for this suggestion. For brevity, we do not make a statement regarding disturbance interactions with drought in the revised version of the manuscript. Hence, inclusion of a reference here is no longer appropriate.

p. 9, Climate data, 1st sentence; what time period you choose for the 30-year normal?

Thank you, the temporal dimensions of the climate data were certainly important in the originally submitted version of the manuscript. However, per reviewer critiques we have removed our analysis of climate effects on species population performance in the revised version of the manuscript.

p. 14, no. 46.; pinon-juniper, but see no. 47. piñon-juniper

This is an excellent catch, thank you. Throughout the text we use “pinyon”, however, we do not believe it is appropriate to modify the titles of our references to match our semantics. Our understanding is that all versions listed above are correct.

Comments by Reviewer #2

General: The manuscript by Stanke et al. is an interesting analysis of changes in species abundance for dominant conifers across the western US. I found the method development to be the most compelling part of this research. The FSI index represents a standardized means for looking at changes in the density and dominance of species within the FIA plot network.

That said, I have serious reservations about the interpretation of the results, the novelty of these findings, and in particular the framing used to convey the findings. At the most basic level, the authors are stating that populations of some of the most dominant species in the western US are in decline due to disturbance. This is not surprising or a novel finding in my view. To support this finding they note that the FSI index calculated from 2001-2018 is negative for many of the conifer species in question. I don't dispute this. My criticism lies in the framing of the problem. The authors tend to equate 'population performance' with abundance. An extensive body of research in western forests has noted that active fire suppression since the 1940s has resulted in current forests with substantially higher densities than the pre-suppression era. Moreover, today's forests have species compositions that are notably different than historical conditions and are often dominated by shade tolerant species at higher proportions than historic conditions. Thus the recent decline in shade tolerant species these authors note needs to be contextualized with respect to historical conditions. Moreover, the period from 1940-1980 was a relatively cool wet period for much of the study domain. The study interval here (2001-2018) represent a period of recent rapid warming and changes in the fire regime of these forests after this period of active fire suppression with mesic conditions. One could argue that a reduction in tree density is actually returning forest density and composition to conditions that were more common in the past which could be considered simply a reversion to historical norms. This study does not provide a means to contextualize the changes in abundance presented and the short time interval of this study (18 years) doesn't provide the context needed to interpret the changes observed. Moreover, the formulation of the FSI index combines both changes in density (TPH) and basal area. I would argue that the loss of large trees is a bigger issue with respect to ecological functioning in these systems compared to changes in TPH per se. There is clear evidence that we are losing large trees in our forests and this has a number of negative implications for the resilience of these systems. The FSI index doesn't allow us to isolate this effect from changes in density. I would find an analysis that focused on large trees to be much more compelling than the one presented here. In sum the framing of this study seems to suggest that greater abundance of trees = greater population performance. This framing may work in eastern forests but the active disturbance regimes in western forests makes abundance a relatively weak metric of forest health and resilience especially when viewed through a short interval of time.

First and foremost, thank you for your thoughtful and thorough review of our manuscript. Your comments and suggestions played a pivotal role in revising and reframing our work. In light of all reviewers comments, we have provided an improved implementation of the Forest Stability Index, modified the scope of our analysis (plus associated complete re-analysis), and made major revisions to all sections of our manuscript. We provide individual responses to each of your specific comments that follow, and offer a response to your general comments here.

You raised concerns regarding the framing of our analysis, specifically with use of the term "population performance" to describe net changes in tree abundance over a relatively short period of time. We recognize and appreciate this concern, and have replaced nearly all instances of "population performance" with "change

in relative tree density” in the revised version of the manuscript. Furthermore, given the short temporal frame of our study and high disturbance activity in the western US during this period, we recognize that observing decline in relative live tree density (as measured by our updated implementation of the FSI) is not inherently surprising at face value. We explicitly state this on lines 173-175 and 255-257 in our revised manuscript. However, we argue that the rate and pattern of shifts in relative tree density that we observe are alarming, often undesirable, and generally inconsistent with reversion of forests toward historic conditions. For example, over the 18-year study period lodgepole pine, Engelmann spruce, and subalpine fir declined in relative density 32%, 21% and 13% across their respective ranges in the western US (tens of millions of hectares). Further, the rate of decline in relative tree density increased with tree size for each of these subalpine species (relative density of the largest 20% of lodgepole pine declined by more than 50% across the species’ range from 2001-2018).

We recognize that the composition, structure, and disturbance regimes of forests in the western US have undoubtedly changed due to forest management (fire suppression) over the last century, and our failure to provide such context was a certain shortcoming of our original manuscript. In the revised manuscript, we have re-written our discussion to provide this context. For specific examples, see lines 161-180, 201-211, 229-237, and 253-258 in the revised manuscript. If it were possible, we would provide historic relative densities of our study species across the western US to help readers contextualize observed changes in the relative density of those species over the last two decades. However, providing such specific data are nearly impossible due to a lack of data. We instead have reframed our discussion to rely on previous work from the region to provide this context whenever possible.

Finally, per the comments and suggestions of all reviewers, we offer an updated implementation of the FSI in our revised manuscript. The newly defined FSI is a measure of change in relative live tree density and is based directly in allometric size-density laws of plant populations. Please see lines 323-363 for an updated description of the FSI. You noted that the previous implementation of the FSI did not allow for the isolation of changes in density and tree size. The improved implementation of the FSI address this issue directly, allowing the FSI to be summarized by tree size-classes to accurately describe changes in the size-structure of forests. Per your comment that you “would find an analysis that focused on large trees to be much more compelling” than the one we previously presented, we have included an additional analysis in our revised manuscript that addresses change in species size-distributions across their ranges in the western US. Please see lines 401-408 in the revised manuscript for a description of this analysis.

Intro paragraph 2 - I appreciate the general notion that we need to understand multiple facets of vital rates to understand population dynamics but what compensatory mechanisms can offset declines in recruitment rates? One may be higher longevity of trees but all evidence as you note seems to point against that including recent work that suggests larger trees are preferentially being lost in forests globally.

Thank you for this comment. We agree that increased longevity of trees is one potential mechanism that could offset decline in tree recruitment and that the global decline of large trees is of grave concern. Per your comments, we have included an analysis of change in relative live tree density across species-size distributions in the revised version of this manuscript. Following such changes in the scope of our analysis, we have modified the introduction of the manuscript considerably and hope to alleviate any concerns you have conveyed here. Furthermore, our updated implementation of the FSI is now sensitive to both changes in absolute stem density *and* tree size distributions. Please see lines 323-363 in the revised manuscript for an updated description of the FSI.

Page 2 paragraph 2 - "For example, elevated population performance in the cooler portions of a species' range relative to warmer portions may indicate species migration along the temperature gradient over time (i.e., shift in species abundance distribution toward cooler regions)." You level criticisms against space-for-time substitutions in the previous paragraph but here you are basically advocating for just that.

We appreciate this comment and agree this is a shortcoming of the previously submitted manuscript. Per reviewer suggestions, we have removed our analysis of climate effects on species abundance shifts. We have revised our introduction to reflect this change and in the process removed the statements regarding climate gradients that you mention here. Additionally, we have refined the scope of our revised manuscript and now place far less emphasis on shifting species distributions. As such, our previous statements offering criticism of space-for-time substitutions have not been included in the revised version of the manuscript.

Second to last paragraph in the intro. In the text here you state the 10 most abundant species but your results are for the 8 most abundant.

Excellent catch, thank you. We have updated the text to the "8 most abundant". Please see line 53 in the revised manuscript.

Figure 2 caption. I would suggest providing a plain language description of what the FSI index means and how to interpret the values shown.?

Thank you for this suggestion. In the revised version of the manuscript we present the FSI in units of % change in relative live tree density to improve interpretation of the index. Furthermore, we have added descriptions of the FSI to each figure caption to improve interpretation of the values. Please see captions of Figures 2-5 for examples of such changes.

Figure 4 caption and throughout text. This term 'range wide' is misleading as it implies declines across the entire range. Figure 3 shows that there is clear spatial variability in FSI with opposing signs of the index. Is there a better way to describe this? Maybe 'range averaged'

This is an excellent suggestion, thank you. In the revised version of the manuscript we have replace all instances of "range-wide" with "range-average."

I find it interesting that Douglas Fir and Ponderosa pine are the two dominant conifers that experienced notable increases in abundance. Doug fir is an extremely plastic species found over an incredible range of environmental conditions. It is also shade tolerant. Im not surprised by its increase in abundance. Ponderosa pine however is surprising given that there are many studies that show it is struggling to naturally regenerate at the hotter drier sites in which it is found. Did you exclude FIA plots with plantings? These two species are the most widely planted post-fire.

Thank you for this comment. In the originally submitted manuscript, plots with evidence of planting were not removed from our analysis. However, in the revised version of the manuscripts we have restricted our analysis to FIA plots with no evidence of forest management (i.e., silvicultural intervention, tree planting, harvesting) within the plot remeasurement interval or preceding 5 years. Please see lines 366-368 for further description of these changes in the revised manuscript.

Figure 5 suggests that Douglas Fir, lodgepole pine, and subalpine fir are all showing improved performance (higher abundance) in hotter and drier areas within their range. As you note this is counter to expectations and other studies that show increased mortality and reduced recruitment rates for these species in hotter drier environments. How do you reconcile your results with this broader literature?

Thank you, this is a certain shortcoming of the originally submitted version of the manuscript. Per reviewer critiques we have removed our analysis of climate effects on species abundance shifts in the revised version of the manuscript.

Page 7 last paragraph. Climate is intrinsically local and the gradients experienced over short distances along an elevational transect are dramatic compared to those experienced over large latitudinal gradients. Is there any elevational patterning in the FSI index?

Thank you for this suggestion. We do not examine patterns in the FSI across elevational gradients in either the original or revised versions of the manuscript, however examination of such patterns would certainly be valuable in future efforts. For the sake of brevity, we do not feel it is appropriate to add such analysis to this manuscript, particularly given that we removed the analysis of climate effects on species population performance in the revised version.

Page 8 paragraph 1. The term "heightened severity" implies that rates are different than previous time intervals...but we don't know that from this study and no citations to supporting work are provided.

We appreciate this comment, and agree this statement requires additional context. At this time it is not feasible to extend our analysis to previous time periods due to a lack of available data (lack of widespread, systematic samples like those available via FIA). Thus in the revised version of the manuscript, we remove the term "heightened severity" when referring to the FSI.

I don't find the section describing how FSI varies with climate compelling. You find a pattern that is counter to a broad array of literature but provide little in the way of why this is the case. My suspicion is that the patterns described with respect to climate are being conflated with the legacy effects of fire suppression and indirect effects of climate on disturbance dynamics. This needs more attention.

Thank you for this comment. It is certainly possible that patterns of species abundance shifts along climatic gradients are an artifact of past forest management and modified disturbance regimes. In light of reviewer comments we have removed such analysis from the revised version of the manuscript.

"The FSI is a measure of population performance". These are loaded terms as I noted at the beginning of this review. Why not call the FSI a measure of species abundance as this is a clearer depiction of what it is.

We appreciate this comment and have replaced nearly all instances of "population performance" with "change in relative tree density" in the revised version of the manuscript.

Methods. You standardize changes in tree density and BA using the stdev of these values across all plots as measured over the study interval. Does this imply that changes in these variables are stationary processes? As I noted, long term exogenous processes like changing disturbance regimes and changing climate will strain a stationarity assumption.

This is an excellent point, although per reviewer suggestions we have provided a new implementation of the FSI in the revised version of the manuscript and the updated implementation of the FSI no longer relies on standard deviation of change components.

Comments by Reviewer #3

General: The study "Over half of most abundant tree species in decline in the western United States" by Stanke et al. presents an analysis of recent demographic changes in the forests of the western US. The authors tackle a highly relevant topic, given forest disturbance regimes are increasing around the globe and have the potential to threaten the integrity and resilience of forest ecosystems. The study is unique in that it relies on a very large observational dataset of repeated measurements, i.e. it reports on actual changes (rather than using the much more widely used space-for-time substitution). Specifically, the authors develop a new index of demographic change in forests, and apply it to the eight most important species of the Western US. They show that a large number of these species are declining. The authors attribute these changes to forest disturbances, mainly wildfire and insect outbreaks. As mentioned the study is highly relevant; it has, however, a number of grave shortcomings in its current form.

First, and most importantly, I find that the key conclusions (regarding the main trend in demographics and their key drivers) are premeditated by the data and analysis performed. The time frame of change analyzed here is 9.78 years (i.e., the mean time span between two inventory censuses), and the data were collected between 2001 and 2018. The authors present a clever new index of demographic change that balances changes in stem density with changes in basal area (more comments on the index below). While I like the overall idea of the index, it can only give negative values (indicating decline) for a period of high disturbance activity (such as 2001 - 2018, which we know from other analyses) and short observational intervals (<10 years), as it takes a forest longer than 10 years to compensate disturbance-induced relative basal area losses through increasing relative stem number via regeneration. If you would be able to allow for a longer regeneration time window, such as 30 years, things might actually become interesting, but this is not yet possible with the currently available data. For the same reason, the change attribution part of the paper is not really meaningful because the index *has* to give negative values with disturbance. Identifying disturbance as the most important driver is thus circular reasoning.

Second, while I generally like the idea of the index and the trade-offs between standardized change in stem number and standardized change in basal area, I find the index to be defined ad hoc, with the authors never actually showing that the underlying premises hold true. In other words: Is demographic constancy really defined by a 1:1 trade-off between standardized basal area change and standardized stem density change? I would think not, because stem number decreases roughly with the 1.605 power of diameter (Reinekes rule) while basal areas increases with the 2.000 power of diameter. So rather than just allude to scaling laws throughout the text I suggest to actually base the index on them, in order to justify the ad hoc assumptions that go into it.

Third, while the authors maintain that their analysis factors out regular transient changes in forest ecosystems, such as succession, I find that I strongly disagree with this notion. The index is indeed clever, but when applied at the level of individual species, such as done here, it does not consider regular successional patterns in forests. When an old-growth system is burned, for instance, it will necessarily lead to a decline in the dominant late-seral species, and no one would expect these species to again gain dominance within 10 years of a disturbance. Rather, early-seral species will take over for some time (decades to centuries), and only gradually the late-seral species that dominated previously will reinvade the area and attain dominance once more. In your analysis this type of succession (which happens regularly in most forests around the world) would

show up as a steep decline in the late-seral species, while it is only a temporal decline that prevails only as long as the early seral stages are maintained.

In addition to these methodological flaws, the text is vague in many respects. It, for instance, completely misses a temporal reference throughout; only in the Methods the time frame of analysis is mentioned. Furthermore, the text talks about population increase and decline, however it is not clear for the reader whether a decline means a reduction in the range of occurrence or a reduction in the population level - two very different things. These aspects should be made clear early in the text in order to avoid confusion for the reader.

Lastly, the study is distinctly local in its framing and contextualization. Despite the fact that the last sentence of the abstract alludes to the global temperate forest, the narrative and references used in the introduction and discussion are purely centered on the specific discussion in the Western US. This might be appropriate for a society journal such as an outlet from the Ecological Society of America, but not for a globally relevant journal from the Nature publishing group. Why would this be relevant for a reader in Asia or Australia, for instance?

One last note before I list a number of more detailed comments on the text: Next time you submit a manuscript, please provide line numbers so that comments can be made in reference to specific lines!

Thank you for your thoughtful and thorough review of our manuscript. Your comments and suggestions were of immense value in revising our work. In light of all reviewers comments and suggestions, we have provided an improved implementation of the Forest Stability Index, modified the scope of our analysis (plus associated complete re-analysis), and made major revisions to all sections of our manuscript. We believe that our revised manuscript has improved by orders of magnitude relative to the originally submitted version. We sincerely apologize for the lack of line numbers in our original manuscript, we have addressed this issue in our revisions. We provide individual responses to each of your specific comments that follow, and offer a response to your general comments here.

First, and most importantly, you raised concerns regarding the key conclusions presented in our original manuscript being premeditated by the data and analysis performed. Specifically, you note that our previous definition of the FSI *has* to give negative values during a period of high disturbance activity. Under the revised definition of the FSI (we talk more about these changes below) this statement is not true, particularly when summarized at the landscape level. At the stand-scale, when disturbance occurs within a short observation interval a decrease in relative live tree density and hence a negative FSI will almost certainly result. However, at the landscape level, stands where relative density declines due to disturbance will be offset by increases in relative density in stands responding to *previous* disturbance (e.g., pure recruitment) assuming temporally constant disturbance severity and disturbance probability (i.e., landscape equilibrium). All results in revised manuscript are presented at the landscape/regional scale. Hence, we argue that the results and key conclusions presented in our revised manuscript are *not* premeditated by our analysis. If disturbance severity and/or probability have changed during the study period, we would expect to detect this with the FSI. The primary purpose of this study is quantify the net consequences of such changes for a suite of important temperate tree species.

Furthermore, our revised definition of the FSI does not have to give negative values at the stand-scale when summarized by species over a short period of time (10 years, as used here). For example, consider a simple two aged-stand with a ponderosa pine overstory and Douglas-fir understory (advance regeneration). In the event that a disturbance agent selectively results in mortality of large-diameter overstory trees (e.g., mountain pine beetle outbreak), we would expect to see a decline in relative density of ponderosa pine due to elevated mortality and a shift in the species' size distribution toward smaller-diameter stems. However, it is very reasonable to expect Douglas-fir advance regeneration to quickly respond to increased light availability via growth. In this case, the size distribution of Douglas-fir is expected to shift rightward (increase) and thus result in an *increase* in the species' relative tree density in the post-disturbance stand (positive FSI). In short,

when summarized at the species-level, the revised definition of the FSI is capable of detecting inter-specific compensatory responses to disturbance (see response of Engelmann spruce and lodgepole pine to disease in Figure 5).

Regardless, we recognize that broad-scale decline in relative live tree density (as measured by our updated implementation of the FSI) may not, at face-value, be inherently surprising. We explicitly state this on lines 73-175 and 255-257 in our revised manuscript. However, we argue that the rate and pattern of shifts in relative tree density that we observe are alarming, often undesirable, and generally inconsistent with reversion of forests toward historic conditions. We understand and appreciate your concern given the original presentation of our results, and have made a targeted effort in our revisions to alleviate these issues. Please see lines 61-180, 201-211, 229-237, and 253-258 in the revised manuscript for specific examples of these revisions.

We agree that our previous statement identifying disturbance as the most important driver of tree population decline was flawed, and we present an improved analysis of disturbance effects on species relative density shifts in our revised manuscript. Please see lines 420-438 for an updated description of this analysis. In short, we now estimate the average severity (mean difference in FSI between undisturbed and disturbed sites) and frequency (annual probability) of three primary disturbance types within the range of each study species over period 2001-2018. To quantify the relative importance of each disturbance type in driving shifts in relative density of each species, we take the product of disturbance severity and probability. Here, severity times probability yields an estimate of the average annual change in the FSI resulting from each disturbance type, or the *sensitivity* of tree populations to each disturbance type over the study interval.

Per your suggestions, we offer an updated implementation of the FSI in our revised manuscript. The newly defined FSI is a measure of change in relative live tree density and is based directly in allometric size-density laws of plant populations. Please see lines 323-363 for an updated description of the FSI. As you note, we incorrectly stated that the FSI is “independent of successional processes” on multiple occasions in the original version of the manuscript. We have removed such statements from the revised manuscript. We are fully aware of the importance successional processes in driving tree population dynamics, particularly at the level of individual species. In fact, the primary goal of this study is to quantify the net result of disturbance, succession, and other exogenous and endogenous processes on tree population dynamics across broad spatial domains. We do, however, argue that the FSI is independent of transient dynamics in tree demography arising from *stand structural development* (allometric size-density relationships, self-thinning). We provide an ecological basis for this independence in our definition of the FSI on lines 323-363 in the revised manuscript.

Finally, you note that the original version of our manuscript was “distinctly local in its framing and contextualization.” We agree, and recognize this as major shortcoming of the original manuscript. In the revised version of the manuscript, we argue our results are significant for other regions of the temperate biome, and have re-written our introduction, discussion, and conclusion to make this clear. Specifically, we broaden our references to provide support for statements across temperate forests attempt to explicitly mention the relevance of our work in other regions of the globe. Please see lines 14-17 of the revised manuscript for a specific example.

```
p1: individual-tree demographic responses... unclear, as far as I can see
demographic responses are at the level of a population, but not at the level of
an individual tree
```

Thank you for this comment. We agree that demographic responses occur at the population-level rather than the level of the individual. We have updated all instances of “individual-tree demographic response” in the revised version of the manuscript to reflect this.

```
p2: importance of endogenous processes during maturation... I very much agree
here - one important process that is grossly neglected is succession, i.e. any
reasonable forest ecologist would not expect the same species to dominate in an
early-seral patch vs. an old-growth patch! Your analysis also does not account
for this process - see my general comment #3 above.
```

Thank you for this comment, we incorrectly stated that the FSI is “independent of successional processes” on multiple occasions in the original version of the manuscript. We have removed such statements from the revised manuscript. However, we disagree that succession is neglected in this study. Tree population dynamics arise from complex interactions among exogenous and endogenous processes, including succession. In this study, we seek to measure the net result of such processes in influencing tree populations across broad spatial scales. Thus, while we do not explicitly acknowledge succession in our analyses, the effects of succession are undoubtedly manifested in our results. This is our intention...to measure the *net* result of succession, disturbance, and other processes on the population dynamics of tree species.

p2: differ among the 10 most abundant tree species in the western US... in the results section it seems that you only focus on the 8 most abundant species

Excellent catch, thank you. We have updated the text to the “8 most abundant.” Please see line 53 in the revised manuscript.

p2: estimated total number of live stems... I don't think that selecting the most important ones purely based on stem density is meaningful. The reason is that this will strongly bias towards early seral species (generally high densities) over late-seral species (generally low densities). I would suggest that you need to at least also include some measure of dimension, e.g., basal area would be an integrated measure over density and size. As it is, the measure of importance (purely based on stem density) is not consistent with the stability index calculated (based on stem density *and* basal area).

We appreciate this suggestion. However, we have found that defining abundance in terms of basal area rather than stem density has very little effect on the resulting top 8 species. Specifically, when we rank by total basal area the only change to the resulting species list is the replacement of quaking aspen with western hemlock. While we recognize that western hemlock is an important late-seral species in the Pacific Northwest and portions of the Northern Rockies, its range is severely limited in the western US relative to quaking aspen. That is, aspen is far more broadly distributed across our study region than western hemlock (constrained to PNW and Northern Idaho), and thus we feel aspen better represents the western US in its totality than does western hemlock.

Further, our revised manuscript offers an updated implementation of the FSI that effectively measures temporal change in the relative density of live trees (please see lines 323-363 for a revised description). Indices of tree size are, of course, still critical in determining relative tree density. However, we feel the interpretation of the revised FSI is much more consistent with ranking species' abundance by stem density as opposed to basal area or areal prevalence.

p3: Tendency towards range-wide population decline occurred more frequently... you talk about decline and expansion here, but it is not clear for the reader at this point what your temporal reference is; are we talking about the last five years, the last 50 years, the last 500 years here? Would be important to know to interpret your results!

Thank you for this comment, we have re-written all sections of our manuscript in an effort to clarify the temporal reference of our analyses.

Figure 2: You talk about confidence intervals in the figure caption, but I can't see them in the figure.

This is an excellent point. In each figure where confidence intervals are mentioned (Figures 2-3), the 95% confidence intervals associated with point estimates fall within the width of the “dots” in the Figure. In other words, confidence intervals are smaller than can be effectively shown in the figure (extremely large sample sizes help improve precision). Thus, we instead emphasize the interpretation of the confidence intervals by coloring “dots” as significant (red or blue, excluding zero) or not-significant (grey, not excluding zero).

Figure 2: Give an example in the figure caption what a FSI of -0.2 means. As it is the numbers are very hard to contextualize for the reader.

Thank you for this suggestion. In the revised version of the manuscript we present the FSI in units of % change in relative live tree density to improve interpretation of the index. Furthermore, we have added descriptions of the FSI to each figure caption to improve interpretation of the values. Please see captions of Figures 2-5 for examples of such changes.

p3: a negative relationship between the FSI and forest disturbance severity for all species and disturbance types... this is not a result, this is a foregone conclusion based on how you defined your FSI. From what I can see it is technically not possible that a disturbance results in stable/ positive FSI responses. So the analysis is circular. See also my general comment above.

We appreciate this comment, and agree that this portion of our original analysis was flawed. In our revised manuscript, we offer an improved analysis that we believe will alleviate the concerns you express here. For further detail, please see our response to your general comments above (where this issue was first raised) and lines 420-483 in the revised manuscript (where we describe our improved analysis).

p3: fire emerged as the most important predictor for all species except lodgepole pine... again, these statements lack a temporal reference. Fire when? In the year prior to the last observation? 20 years ago? 200 years ago? These forests are fire-driven systems, so the fact that fire resets forest dynamics and instigates regeneration is a part of their natural dynamics.

Thank you for this comment. We have re-written all sections of our manuscript in an effort to clarify the temporal reference of our analyses, and offer an improved analysis of disturbance effects on species relative density shifts (please see general comments and lines 420-483 in the revised manuscript).

p4: in historically warmer portions of the species' ranges... what is historically warmer portions, which part of history are you referring to? Last year? The Holocene? And are there really portions of your study region that were warmer than they are currently in the last, say, 20 years?

Thank you, the temporal dimensions of the climate data were certainly important in the originally submitted version of the manuscript. However, per reviewer critiques we have removed our analysis of climate effects on species population performance in the revised version of the manuscript.

Figure 3: ecoregion subsection (top) and ecoregion division... I find the two aggregation levels are not needed - they show the same information, only at different grain. Go for the finer grain (and put the coarser grain into the supplement, if needed).

We appreciate this suggestion, and agree that the two levels of aggregation sometimes show similar patterns at different grain. However, we (and Reviewer 1) feel that displaying 3 levels of aggregation in the main text (i.e., range-average, ecoregion division, and ecoregion subsection) offers a unique opportunity to examine how patterns of relative density shifts vary with respect to spatial scale.

For example, subsection-level estimates of the FSI for Douglas-fir in the Rocky Mountains and interior PNW show a high degree of variability across relatively small spatial scales (Figure 4). However when summarized to the division-level, the net result of such local variability is an effectively “stable” population (grey). In our opinion, allowing the reader to easily examine such patterns across spatial scales is of unique value, as it allows the reader to make inferences regarding the spatial scale(s) that primary drivers of tree population dynamics have operated on short time-intervals.

Figure 3: It is unclear how you aggregated the inventory plots to ecoregions. You talk about moving window aggregation in the methods, but I don't understand how you get from moving window values to ecoregions.

Thank you for this comment, we have revised our methods section to clarify this. We use a simple moving average estimator (as described by the FIA program and implemented in the rFIA R package) to estimate the mean and variance of the FSI for each species and within each areal domain (ecoregion subsection, division, or full study region). The interpretation here is very similar to those produced by estimators under simple random sampling. That is, we are simply estimating the mean and variance of the FSI from the sample of FIA plots within our “population of interest” defined by species and spatial domains. Please see lines 390-397 and 413-414 in the revised manuscript for examples of our clarification in the text.

Figure 3, caption: here and throughout the text it is not clear if decline and expand means in area, or in terms of demographic indicators (e.g., fewer stems). These are two different signals, and it should be clear what is reported here.

Thank you, we agree this was certainly a short-coming of the original manuscript. In our revised analysis we treat changes in species' ranges and within range density shifts as functionally equivalent processes (please see lines 83-87 in the revised manuscript). For example, range expansion (detection of a species on an FIA plot where it was previously absent) is represented by the FSI as a positive change in relative density as previous relative density is 0 and current relative density is a positive value. We feel that separating species relative density shifts into change components (i.e., range dynamics and/or within-range shifts) would be very useful and interesting in future efforts. Though in this study, we seek to quantify the net result of these two processes, or the net “performance” of tree populations over a short window of time. Hence, we feel it is appropriate to treat changes in species' ranges and within range density shifts as equivalent processes herein.

p4: Our results indicate a majority (5 of 8) of the most abundant tree species in the western US are undergoing significant range-wide population decline... I suggest to not base your conclusions on majority votes, but on something like % area experiencing decline/ expansion. This is much more robust/ relevant.

We appreciate this suggestion and agree this was a shortcoming of the original manuscript. In the revised manuscript we present this result as % of species in decline (same as original) and the % of stems represented by these 5 species (which is still more than half, 60.7%). Please see lines 161-162 in the revised manuscript.

p7: and is independent of transient dynamics associated with stand structural development... how so? I cannot see how the analysis would not reflect transient successional dynamics of a forest. See my general comment above.

Thank you for this comment, as mentioned above we incorrectly stated that the FSI is “independent of successional processes” on multiple occasions in the original version of the manuscript. We have removed such statements from the revised manuscript. We do, however, argue that the FSI is independent of transient dynamics in tree demography arising from allometric size-density relationships in forests (self-thinning, stand structural development). We provide an ecological basis for this independence in our definition of the FSI on lines 323-363 in the revised manuscript. Furthermore, please see our response to your general comments.

p8: Among the primary values of the FSI is its independence of transient dynamics associated with stand structural development and succession... this is a strong claim that you make repeatedly throughout the text, but I cannot see how this claim can be substantiated. If possible, please demonstrate that the index is independent of succession, e.g. based on an observed (or simulated) successional trajectory that is otherwise stable over time.

Thank you. Please see our above response.

p10: denotes the standard deviation of the variable x across all plots sampled... sd between plots (in space) or sd over time per plot, or both? This matters a lot for the interpretation of your index, and is unclear.

This is an excellent point, although per reviewer suggestions we have provided a new implementation of the FSI in the revised version of the manuscript and the updated implementation of the FSI no longer relies on standard deviation of change components.

REVIEWER COMMENTS

Reviewer #1 (Remarks to the Author):

Dear authors,

Many thanks to all your questions and I am completely fine with them. After careful reading of all comments or the authors' responses to the previous manuscript, I have found that all comments have been answered adequately, all ambiguities have been cleared up and also all minor errors have been corrected, so that I have no further critical comments to make on the revised manuscript. Overall, the content is now clearer, because some corrections were carried out in the data analysis, so that the core statements are more prominent and I consider the improved FSI to be a very good basis for application in other regions.

Reviewer #2 (Remarks to the Author):

This is a substantially improved submission compared to the previous draft I reviewed. I appreciate the effort the authors took to squarely address my comments and those of the other reviewers. I found that the authors have elegantly handled those comments and incorporated the suggested changes in the revised analysis. In particular, I am pleased to see the treatment of the relative density changes broken down by size class distribution and an acknowledgment of the role of previous and ongoing management (e.g. fire suppression) and its effect on the results. I also think it was a strategic decision to drop the analysis of climate in this study as the results are more clear and impactful as they currently stand. My only remaining quibble would be that I think the manuscript would benefit from a clear statement in the discussion addressing the short time frame of this study with respect to the long time frames in which compositional and structural changes may be manifest in long lived forests in the western US. I see the patterns elucidated in this study as capturing short term and maybe transient dynamics in some cases. The extent to which these translate into longer term trends is not entirely clear but this opens up active areas of future research.

I think this is a great contribution to the broader literature in forest ecology and forest biogeography. The study has broad international relevance, and I think this will spur a lot of future work. Thank you for the opportunity to review this.

Solomon Dobrowski

Reviewer #3 (Remarks to the Author):

This is the second time I am reviewing the submission “Over half of most abundant tree species in decline in the western United States” by Stanke and co-workers. In addition to many other comments I had major methodological concerns in the previous round, and will focus exclusively on these methodological issues in my second evaluation. The authors have now considerably revised their methodological approach – in fact the current manuscript uses an entirely different method (based solely on stem density changes, whereas the original submission was based on the combined changes in stem density and basal area). The new approach alleviates a number of problems with the original submission, such as the issue that the index by design almost certainly had to return negative results. Now three of the eight species investigated are in fact showing positive responses (Fig. 2).

Unfortunately, however, the new method remains fundamentally flawed in at least two important ways. First, the authors base their new relative density index on the relationship between stand density (N) and stand basal area (S) – see Figure 6. However, this relationship is spurious, as per definition S is a function of N , specifically S is calculated as $S = N \cdot dbh^2 \cdot \pi() / 4$ from the underlying raw FIA data. So the relationship the index is based on does not make sense. Second, while the authors claim that they use a well-established size-density relationship (1157) they in fact do not – they rather derive their own relationship from quantile regression based on the underlying data. This ultimately leads to circular reasoning: The extreme values in the dataset (i.e., those that have the strongest leverage on the quantile regression the authors use) are the result of recent disturbances. Yet, disturbances are also the subsequent focus of the analysis of the authors. While the authors aim to analyze if/ how disturbances influence populations based on their index, this is not possible because the index has the recent disturbances hardwired into its core.

Statement on the Review of
⟨NCOMMS-20-20430⟩
based on Referee's Report

Hunter Stanke, Andrew O. Finley, Grant M. Domke, Aaron S. Weed, David W. Macfarlane

October 21, 2020

This statement concerns our second revision of the ⟨NCOMMS-20-20430-A⟩ paper, entitled “*Over half of most abundant tree species in decline in the western United States,*” based on the referees' report. All reviewer comments have been listed in their original form (black text), along with individual responses to each comment (blue text). In all responses that follow, line numbers refer to those in the revised manuscript where appropriate modifications have been made.

Comments by Reviewer #1

Dear authors, Many thanks to all your questions and I am completely fine with them. After careful reading of all comments or the authors' responses to the previous manuscript, I have found that all comments have been answered adequately, all ambiguities have been cleared up and also all minor errors have been corrected, so that I have no further critical comments to make on the revised manuscript. Overall, the content is now clearer, because some corrections were carried out in the data analysis, so that the core statements are more prominent and I consider the improved FSI to be a very good basis for application in other regions.

Thank you for your kind words and thorough review of our work. We very much appreciate your time and efforts in reviewing our revised manuscript.

Comments by Reviewer #2

This is a substantially improved submission compared to the previous draft I reviewed. I appreciate the effort the authors took to squarely address my comments and those of the other reviewers. I found that the authors have elegantly handled those comments and incorporated the suggested changes in the revised analysis. In particular, I am pleased to see the treatment of the relative density changes broken down by size class distribution and an acknowledgment of the role of previous and ongoing management (e.g. fire suppression) and its effect on the results. I also think it was a strategic decision to drop the analysis of climate in this study as the results are more clear and impactful as they currently stand. My only remaining quibble would be that I think the manuscript would benefit from a clear statement in the discussion addressing the short time frame of this study with respect to the long time frames in which compositional and structural changes may be manifest in long lived forests in the western US. I see the patterns elucidated in this study as capturing short term and maybe transient dynamics in some cases. The extent to which these translate into longer term trends is not entirely clear but this opens up active areas of future research.

I think this is a great contribution to the broader literature in forest ecology and forest biogeography. The study has broad international relevance, and I think this will spur a lot of future work. Thank you for the opportunity to review this.

First and foremost, thank you for your kind response and thoughtful review of our revised manuscript. We genuinely appreciate your time and suggestions. We agree that the manuscript would benefit from a clear statement that addresses the temporal frame of our study, and have added such a statement in the concluding paragraph of our discussion. Please see lines 305-309. Thank you again for your efforts in reviewing our work!

Comments by Reviewer #3

This is the second time I am reviewing the submission ‘‘Over half of most abundant tree species in decline in the western United States’’ by Stanke and co-workers. In addition to many other comments I had major methodological concerns in the previous round, and will focus exclusively on these methodological issues in my second evaluation. The authors have now considerably revised their methodological approach - in fact the current manuscript uses an entirely different method (based solely on stem density changes, whereas the original submission was based on the combined changes in stem density and basal area). The new approach alleviates a number of problems with the original submission, such as the issue that the index by design almost certainly had to return negative results. Now three of the eight species investigated are in fact showing positive responses (Fig. 2).

Unfortunately, however, the new method remains fundamentally flawed in at least two important ways. First, the authors base their new relative density index on the relationship between stand density (N) and stand basal area (S) - see Figure 6. However, this relationship is spurious, as per definition S is a function of N , specifically S is calculated as $S = N * dbh^2 * \pi() / 4$ from the underlying raw FIA data. So the relationship the index is based on does not make sense. Second, while the authors claim that they use a well-established size-density relationship (1157) they in fact do not - they rather derive their own relationship from quantile regression based on the underlying data. This ultimately leads to circular reasoning: The extreme values in the dataset (i.e., those that have the strongest leverage on the quantile regression the authors use) are the result of recent disturbances. Yet, disturbances are also the subsequent focus of the analysis of the authors. While the authors aim to analyze if/ how disturbances influence populations based on their index, this is not possible because the index has the recent disturbances hardwired into its core.

Thank you once again for reviewing our manuscript and providing a thoughtful and thorough response. We very much appreciate your time and efforts. You provide two distinct critiques of the revised FSI, however it appears both arise from a misunderstanding of one of the key variables underlying the index: **average tree basal area** (S), an index of average tree size that is independent of stand density (N). Furthermore, you suggest our reasoning is ‘‘circular’’ and that recent disturbances are ‘‘hardwired’’ into the core of the FSI. We disagree, and we suspect you missed our statement on line 395-397 stating that recently disturbed plots were removed prior to fitting maximum size-density models. Please see our response below for a more complete rebuttal.

In your response, you suggest that our index of relative tree density is based on a relationship between stand density and **stand basal area**. Unfortunately this is not true, and we believe the misunderstanding may have originated from the axis label in Figure 6 given that you reference this figure in your response. Rather, our index of relative tree density is based on well established relationships between stand density and **average tree basal area**, or the size of the average tree within a stand. Average tree basal area is equivalent to stand basal area divided by stand density and is directly related to quadratic mean diameter (as used in Reineke’s stand density index).

You are correct in that stand basal area is a function of stand density, and that an index of relative tree density based on stand density and total stand basal area (as described by the equation you provide) would be fundamentally flawed. However, this is simply not the case in our index and we sincerely apologize for this confusion. In the revised version of our manuscript, we have updated the x-axis label in Figure 6 to state ‘‘average tree basal area’’ instead of ‘‘basal area,’’ updated line 344 to explicitly state ‘‘average tree basal area,’’ and updated line 394 to show that average tree basal area (\overline{BA}) is computed as tree basal area per hectare (i.e., what you refer to as stand basal area here) divided by trees per hectare within a stand. In addition, please refer to caption of Figure 6 and the lines 342, 344, 356, and 394 in the manuscript where we explicitly define S as an index of tree size/ basal area (where \overline{S} denotes average tree size/ basal area).

Second, you suggest that we do not base our index in well-established allometric size-density laws. If

our index was computed from stand density and stand basal area, as you suggest, this would certainly be true. Though again, our index of relative tree density (and subsequently the FSI) is based on indices of tree density and tree size (i.e., basal area). Allometric relationships between plant size and density are widely accepted as quantitative law describing the behavior of plant populations under “self-thinning” conditions (please see lines 347-349 of our manuscript and the references therein). You explicitly reference “Reineke’s rule” (otherwise known as Reineke’s law of self-thinning) in your response to our original submission, which arises from some of the earliest work on this exact topic. Specifically, Reineke showed that maximum tree density declines with increasing quadratic mean diameter in even-aged forests, and that such decline follows a power-law. Reineke’s seminal paper (<https://naldc.nal.usda.gov/download/IND43968212/PDF>), and nearly all work that followed, acknowledges that the *coefficients* of the power function (at least one of the two) vary by species and ecological setting, although the functional form of the relationship (i.e., a power law) remains the same. As such coefficients are not available for all forest types observed in the western US, *they must be estimated from our underlying data* to derive the maximum size-density curves that are used in nearly all indices of relative tree density. Our index differs from Reineke’s original “Stand Density Index” in a number of key ways (see lines 354-372), though we use the same index of tree density (i.e., tree number per unit area) and effectively the same measure of tree size (i.e., average tree basal area can be derived exactly from quadratic mean diameter: $\overline{BA} = \pi(0.5 * QMD)^2$). Again, we suspect your response was based on the misunderstanding that we used stand basal area in place of average tree basal area in our analyses, and genuinely appreciate your comments here.

Importantly, you further suggest that our reasoning is circular because we estimate maximum size-density relationships (i.e., reference curves) with the same data that are used in our subsequent analysis of disturbance effects on tree populations. We disagree with this statement for two key reasons. First, as stated in lines 395-397, *we removed all recently disturbed plots prior to fitting our models* of maximum size-density curves. Second, while we agree that the inclusion of recently disturbed plots will affect the coefficients estimated via quantile regression (hence we removed them), we disagree that recently disturbed plots exhibit the greatest “leverage” in the regression. In general, recently disturbed plots are likely to fall in the lower half of the tree density distribution (i.e., lower 50th percentile) because disturbances act as “thinning agents” in tree populations. However, we are attempting to model the *maximum* size-density relationship, i.e., the 99th percentile of the tree density distribution conditional on tree size. Therefore, we would argue that recently disturbed plots are likely to exhibit the *least* leverage in the regression, though they’re inclusion would almost certainly shift the 99th percentile of the tree density distribution downward (imagine adding a large probability mass at low tree densities) and subsequently cause us to underestimate the maximum size-density curve. For the sake of example, let’s consider the alternative scenario in which recent stand-replacing disturbance incites very rapid (within the 10yr remeasurement interval) tree regeneration (very uncommon as you point out in your response to our original submission). What we would have is a dense stand with a small average tree size, i.e., the plot would fall in the upper left of Figure 6. We can think of no ecological basis to justify excluding such a plot from our analysis. In fact, the plot would likely contribute important information regarding the range of potential stand densities that could be observed in a stand with a very small average tree size (again this is an uncommon scenario given our definition of recent disturbance - occurring during the 10 year remeasurement interval or 5 years preceding).

Finally, you state that it is not possible to examine the relative influence of disturbance processes on tree populations using the FSI, because the index has “recent disturbances hardwired into its core”. As discussed above, we expect you missed our statement on lines 395-397 stating that recently disturbed plots were removed prior to fitting maximum size-density models. For the sake of argument, however, let’s consider an alternative scenario where we don’t remove such plots. That is, we fit our regression using measures of tree size and density at *all* of our plot locations. Our model would almost certainly be biased (i.e., we would underestimate maximum size-density curves), and subsequently we would tend to *overestimate* relative tree density for each plot visit (as relative tree density is observed density divided by maximum density). The FSI is defined as the annualized difference in relative tree density observed in a stand over time. If we assume that bias in the maximum size density curve is not conditional on tree size (i.e., equal bias across all tree sizes), then such bias is “subtracted out” in the computation of the FSI. Let’s say this is not the case, and that bias is greater in small tree size classes than large tree size classes (i.e., we underestimate the maximum size-density curve more for small diameter stands than large ones). On average, disturbance will reduce both tree density and average tree size, and hence the FSI will generally be negative when disturbance occurs

on a plot (i.e., relative tree density declines). In this case, bias in our stand density curve will cause us to *underestimate* average disturbance severity because we are more likely to overestimate relative density in small diameter stands (see lines 438-460). Annual probability of disturbance, the second component of our disturbance analysis, is obviously independent of our size-density curves. Thus, while we agree that including recently disturbed plots in estimation of maximum size-density curves (which we *did not do*) may bias an assessment of the influence of disturbance processes on tree populations using the FSI, we find no basis upon which such an analysis could be considered “circular” or in which recent disturbance are “hardwired” into the computation of the Forest Stability Index.

Again, our sincere thanks for providing a second review of our manuscript. Your comments and suggestions have been invaluable in improving our work, and we very much appreciate your time.